# UNIFYING MULTI-SCALE DESIGN IN TIME-SERIES FORECASTING

## ABSTRACT

Multi-scale modeling in time-series forecasting, which seeks to capture cross-scale relationships for modeling complex dependencies, is increasingly popular. While previous work lacks principled foundations, we unify existing scaling methods into a *scaling operator family,* providing a general theoretical basis for multi-scaling methods and revealing two key limitations of current models: static scaling and inflexible cross-scale modeling. To address these limitations, we propose SIGMA (Single Gaussian Multi-scale Architecture), a simple yet principled multi-scale framework. It enables position-wise scaling via the learnable discrete Gaussian (LDG) kernel grounded in scale-space theory, coupled with a lightweight MLP processor for efficient cross-scale interaction. We evaluate SIGMA comprehensively on long- and short-term forecasting benchmarks against state-of-the-art multi-scale baselines. SIGMA outperforms all competitors on both tasks, achieving the best performance in 55 out of 80 long-term evaluation settings. Beyond accuracy, SIGMA improves training speed by up to 3.8 times and reduces memory consumption by up to 5.3 times compared to the strongest competitors. Code is available at `https://anonymous.4open.science/r/SiGMA-ICLR2026`.

## 1 INTRODUCTION

Time-series forecasting is widely applied in diverse domains, including energy systems (Deb et al., 2017), climate and weather prediction (Dimri et al., 2020), and traffic management (Tedjopurnomo et al., 2020), to name a few. Despite its importance, forecasting remains fundamentally challenging due to the complex temporal structures of real-world time series (Kolambe, 2024). Inspired by the success of deep learning, recent work has introduced neural forecasting methods, including MLP-based methods (Ekambaram et al., 2023; Zeng et al., 2023; Yi et al., 2023; Challu et al., 2023) and Transformer-based methods (Nie et al., 2023; Liu et al., 2024; Shi et al., 2025).

Within this scope, multi-scale modeling has emerged as a powerful design principle. By constructing temporal dynamics at multiple resolutions, it explicitly models cross-scale interactions, thereby enhancing predictive performance. Specifically, coarse-grained scaling captures long-term dynamics, whereas fine-grained scaling preserves high-frequency fluctuations, yielding a more comprehensive representation (Chen et al., 2023). Building upon this principle, recent work has introduced diverse strategies from simple multi-rate sampling to hierarchical and frequency-aware mechanisms (Zhao et al., 2024; Wang et al., 2024; 2025; Naghashi et al., 2025; Hu et al., 2025a; Yang et al., 2025).

We visualize popular scaling operators in Figure 1. Each scaling operator takes a scale parameter $s$ as an additional input to transform the given sequence, creating a representation with different length and resolution. Thus, the effectiveness of a multi-scale approach depends on not only the choice of the scaling function, but also its scale parameters and how to fuse multi-resolution representations.

However, existing methods exhibit two key limitations. (1) *Static scaling.* Most approaches employ a fixed scaling strategy (Wang et al., 2023; 2024; Hu et al., 2025a) for different time steps. This restricts the diversity of temporal views and limits the ability to capture time-varying characteristics inherent in real-world data. (2) *Inflexible cross-scale modeling*. Cross-scale interactions are modeled in a staged manner, processing each scale independently before combining them (Zhong et al., 2024; Wang et al., 2025). This procedure introduces sequential dependencies, reducing efficiency and limiting flexibility in modeling complex temporal relationships.

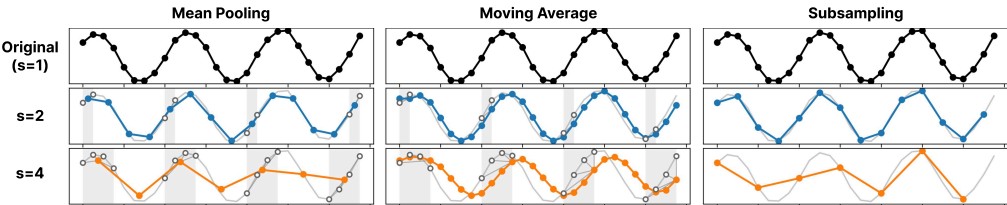

Figure 1: Examples of popular scaling operators used in existing methods. Each operator transforms the input into coarser representations, with a different scaling parameter $s$, capturing different levels of structure and abstraction. Refer to Table 1 for more information of existing scaling operators.

Before presenting our approach to alleviating these limitations, we introduce a novel and principled concept: a *family of scaling operators*, which unifies the scaling methods studied in the literature. We further demonstrate that any multi-scale forecasting model can be decomposed into two essential components: a *multi-scale generator* that creates multiple representations of the input series, and a *multi-scale processor* that fuses information across these representations. This abstraction provides a unified analytical framework for reinterpreting multi-scale time-series modeling.

Based on this unified view, we propose SIGMA (Single Gaussian Multi-scale Architecture), a simple yet principled framework for multi-scale modeling. It adopts position-wise scaling through the learnable discrete Gaussian (LDG) kernel, where time-indexed scale parameters are learned from data within a single operator grounded in scale-space theory. Then, a lightweight multi-layer perceptron (MLP) models cross-scale interactions in a single step, removing staged fusion. Together, these components provide a theoretically grounded and efficient approach that simplifies and strengthens multi-scale modeling.

We evaluate SIGMA extensively on long- and short-term forecasting benchmarks and observe consistent improvements over state-of-the-art multi-scale models. Especially on long-term tasks, SIGMA achieves the best performance in 55 out of 80 evaluation settings with up to 7.8% lower error than runner-up models. Beyond accuracy, SIGMA delivers clear efficiency gains, reducing memory usage by up to 3.8× and increasing training speed by 5.3× over the strongest baselines. We further support these findings with comprehensive ablation and case studies, confirming that SIGMA provides a principled, and efficient framework for multi-scale time-series modeling.

Our contributions are summarized as follows:

- **Unified framework:** We introduce the novel concept of a scaling operator family, which provides a principled foundation for multi-scale time-series modeling. It unifies existing methods under a generator-processor decomposition and clarifies their key limitations.

- **Simplified architecture:** We propose SIGMA, a simple yet principled multi-scale architecture. It enables position-wise scaling and efficient cross-scale fusion through the learnable discrete Gaussian (LDG) kernel with dynamic scaling parameters and a lightweight MLP processor.

- **Empirical validation:** We empirically show that SIGMA achieves state-of-the-art performance on extensive long- and short-term forecasting benchmarks while substantially reducing computational complexity compared to prior multi-scale models. We also conduct various ablation and case studies to provide deeper insights on the successful behavior of SIGMA.

## 2 PROBLEM DEFINITION AND RELATED WORK

**Problem definition** Given a time series, let $\boldsymbol{x} \in \mathbb{R}^L$ denote an input sequence of length $L$ from it, and $\boldsymbol{y} \in \mathbb{R}^T$ denote the corresponding future trajectory of length $T$. We denote the input space as $\mathcal{X} \subset \mathbb{R}^L$ and the target space as $\mathcal{Y} \subset \mathbb{R}^T$. Our objective is to learn a time series forecasting model $h : \mathcal{X} \to \mathcal{Y}$ that maps a past window $\boldsymbol{x}$ to a $T$-step forecast of its future $\boldsymbol{y}$.

**Data assumption** Throughout this work, we focus on non-trivial datasets whose forecastability $\phi$ is strictly less than 1. The forecastability of a time series is defined as one minus the spectral entropy

Table 1: Representative scaling operator families with associated scale parameters $s$. Each family simplifies the sequence in a structured manner as the scale parameter $s$ increases, where $k \in [1, L_s]$ is the element index and $L_s$ denotes the length of the output sequence. For wavelet decomposition, $f(\boldsymbol{x}|s)$ yields the approximation coefficients at level $s$ using the scaling function $\phi$.

| Operator $f$ | Equation | Scale parameter $s$ | $L_s$ | Usage |
|---|---|---|---|---|
| Average pooling | $[f(\boldsymbol{x}|s)]_k = (1/s) \sum_{i=0}^{s-1} x_{ks+i}$ | Window length | $P/s$ | (Wang et al., 2024) |
| Max pooling | $[f(\boldsymbol{x}|s)]_k = \max_{0 \leq i < s} x_{ks+i}$ | Window length | $P/s$ | (Challu et al., 2023) |
| Moving average | $[f(\boldsymbol{x}|s)]_k = (1/s) \sum_{i=0}^{s-1} x_{k-i}$ | Window length | $P$ | (Wang et al., 2023) |
| Subsampling | $[f(\boldsymbol{x}|s)]_k = x_{ks}$ | Stride | $P/s$ | (Liu et al., 2022a) |
| Segmentation | $[f(\boldsymbol{x}|s)]_k = x_k$ | Length divisor | $P/s$ | (Wu et al., 2023) |
| Wavelet decomposition | $[f(\boldsymbol{x}|s)]_k = \langle \boldsymbol{x}, \phi_{s,k} \rangle$ | Decomposition level | $P/2^s$ | (Murad et al., 2025) |

of its Fourier decomposition (Goerg, 2013), formally given as

$$\phi(\boldsymbol{x}) = 1 - \frac{H(\boldsymbol{x})}{\log(2\pi)} \in [0, 1], \quad H(\boldsymbol{x}) = - \int_{-\pi}^{\pi} p_{\boldsymbol{x}}(\omega) \log p_{\boldsymbol{x}}(\omega) \, d\omega, \tag{1}$$

where $p_{\boldsymbol{x}}(\omega)$ denotes the normalized spectral density of $\boldsymbol{x}$ on $[-\pi, \pi]$, characterizing how concentrated the spectrum of $\boldsymbol{x}$ is. Intuitively, larger forecastability $\phi$ indicates a more concentrated spectrum and thus a stronger predictable structure (Wang et al., 2024; 2025). Most real-world datasets including every benchmark used in our experiments satisfy $\phi < 1$ (see Appendix B). This serves as a mild theoretical assumption and does not restrict the practical applicability of our framework.

**Time series forecasting** The field of time series forecasting has seen a significant evolution from traditional statistical methods to modern deep learning architectures. While statistical models like ARIMA (Box et al., 2015) and state-space models (Durbin & Koopman, 2012) are still used, they are often limited in their ability to capture complex nonlinear and long-range dependencies (Zhou et al., 2021). Among deep forecasting models that learn representations directly from raw sequences, multi-layer perceptron architectures have recently been recognized for their effectiveness and efficiency (Ekambaram et al., 2023; Zeng et al., 2023; Yi et al., 2023; Challu et al., 2023). Concurrently, Transformer-based approaches have demonstrated strong empirical results, particularly for long-horizon forecasting tasks (Nie et al., 2023; Liu et al., 2024; Shi et al., 2025).

**Multi-scale design in time series** Time series often exhibit complex patterns across diverse temporal scales, from rapid fluctuations to long-term trends. To capture such heterogeneity, recent work incorporates explicit multi-scale mechanisms to disentangle and integrate temporal dependencies. These methods include hierarchical decompositions that construct multi-resolution representations via downsampling (Liu et al., 2022b; Challu et al., 2023), frequency-domain or wavelet-based analyses that emphasize periodic structure through spectral transforms (Wu et al., 2023; Cai et al., 2024; Murad et al., 2025), and aggregation schemes that fuse representations across scales (Wang et al., 2023; 2024; 2025). However, existing multi-scale methods rely on fixed scaling and staged cross-scale fusion, limiting adaptability to diverse temporal patterns and flexibility in modeling cross-scale interactions. This highlights the need for more flexible multi-scale modeling.

## 3 FOUNDATIONS OF SCALING IN TIME SERIES

In time series analysis, *scaling* denotes transformations that change the temporal or frequency resolution of data. As illustrated in Figure 1, various operations such as downsampling and moving averages are commonly employed in previous work. However, these techniques have largely been treated as ad-hoc design choices, lacking a unifying theoretical principle. We address this gap by introducing a rigorous definition of scaling operator families. To the best of our knowledge, this is the first work that establishes a unified mathematical foundation for analyzing scaling operations in time series. Detailed proofs of the theorems are provided in Appendix A.

**Definition 3.1** (Scaling operator family). *A set of transformations $\mathcal{F} = \{f(\boldsymbol{x}|s) \mid s \in \mathbb{Z}_+\}$ with $f(\cdot|s) : \mathcal{X} \to \mathcal{X}_s \subset \mathbb{R}^{L_s}$ conditioned on $s \in \mathbb{Z}_+$ is a scaling operator family if it satisfies*

- *Non-expansiveness: For $s_i \in \mathbb{Z}_+$ and $\boldsymbol{x}, \boldsymbol{x}' \in \mathcal{X}$, $\|f(\boldsymbol{x}|s_i) - f(\boldsymbol{x}'|s_i)\|_2 \leq \|\boldsymbol{x} - \boldsymbol{x}'\|_2$.*
- *Energy reduction: For any $s_i, s_j \in \mathbb{Z}_+$ such that $s_i = m \cdot s_j$ for some $m > 1$, $\|f(\boldsymbol{x}|s_i)\|_2 \leq \|f(\boldsymbol{x}|s_j)\|_2$ for all $\boldsymbol{x} \in \mathcal{X}$ and $\|f(\boldsymbol{x}'|s_i)\|_2 < \|f(\boldsymbol{x}'|s_j)\|_2$ for some $\boldsymbol{x}' \in \mathcal{X}$.*

*The scale parameter $s$ determines the level of scaling, and also the output dimension $L_s$.*

**Theorem 3.2.** *Common sequence operations such as max/mean/min-pooling (Zheng et al., 2014), moving averages (Box et al., 2015), subsampling (Hannan, 2009), segmentation (Keogh et al., 2004) and wavelet decompositions (Percival & Walden, 2000) are scaling operator families.*

Definition 3.1 formalizes the expected behavior of scaling. Scaling operators reduce or at least preserve the distances between the resulting sequences, as they are designed to capture diverse properties from a sequence but without adding external information. At the same time, the *energy reduction* property formalizes the intuition that time series with coarser scales should be simpler than finer ones along multiplicative chains of scale parameters: as the scale parameter $s$ increases, the result exhibits less overall energy, with small fluctuations being smoothed out.

Table 1 summarizes the operator families given in Theorem 3.2 with their associated scale parameters. These operators can be grouped into two categories based on how they transform the given sequence: *(i) smoothing operators*, which attenuate fine-grained variability by summarizing adjacent observations (e.g., pooling and moving average), and *(ii) structural operators*, which progressively decompose or resample the signals by enforcing a specific structure (e.g., subsampling and wavelet decomposition). Previous works have typically used the term *scaling* only for one group, but our unified Definition 3.1 suggests that both categories of operators can be understood in the same lens in terms of the non-expansiveness and energy reduction properties.

To empirically verify that the operator families in Table 1 satisfy the requirements of Definition 3.1, we evaluate their average pairwise contraction and induced average energy on the Traffic dataset. As shown in Figure 2, all operator families exhibit strictly smaller output differences than input differences, confirming non-expansiveness, and their energies decrease monotonically over multiplicative scales, confirming energy reduction. These observations demonstrate that the operator families in Table 1 satisfy both conditions of Definition 3.1 on *real-world datasets*. Additional experiments on other datasets are provided in Appendix C.

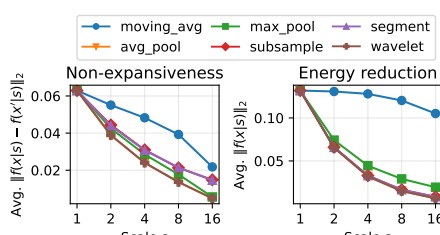

Figure 2: The non-expansiveness and energy reduction of the six scaling operator families on the Traffic dataset.

**Theorem 3.3.** *Trivial operations such as constant mappings, permutations, additive shifts, scalar multiplications, and general linear transformations do not form scaling operator families.*

Theorem 3.3 highlights that not all parameterized transformations qualify as scaling operator families. For example, while a scalar multiplication can change the scale of each observation, it does not guarantee energy reduction and thus falls outside our definition. Putting Theorem 3.2 and Theorem 3.3 together, we claim that our definition of scaling operator families is designed carefully to include only the operations that have been considered as "scaling" in the literature, to provide more interpretable and theoretically grounded approaches to multi-scale time series modeling.

## 4 REVISITING MULTI-SCALE METHODS IN TIME-SERIES FORECASTING

Based on the definition of scaling operator families, we unify existing multi-scale time series forecasting methods and analyze their limitations from the perspective of unified modules.

### 4.1 GENERALIZATION OF MULTI-SCALE METHODS

We formally define multi-scale time series modeling as the composition of two fundamental components: *multi-scale generation* and *multi-scale processing*. For an input sequence $\boldsymbol{x} \in \mathbb{R}^L$, scaling parameters $\boldsymbol{s} = (s_1, \ldots, s_M)$ where $M$ is the number of scaling parameters, and a scaling operator family $\mathcal{F}$, a *multi-scale generator* $g$ is defined as

$$g(\boldsymbol{x}|\boldsymbol{s}) = (f(\boldsymbol{x}|s_1), \cdots, f(\boldsymbol{x}|s_M)),
\qquad (2)$$

which produces a collection of representations of $\boldsymbol{x}$ across the scaling parameters in $\boldsymbol{s}$.

A *multi-scale processor* $p_\theta$ is a function that integrates the representations $g(\boldsymbol{x}|\boldsymbol{s})$ to capture meaningful cross-scale interactions with its learnable parameters $\theta$. For example, some approaches employ weighted summation and aggregation modules to integrate information (Wu et al., 2023; Cai et al., 2024), while others adopt cross-scale mixing mechanisms that dynamically enable interaction across temporal resolutions (Hu et al., 2025a; Wang et al., 2024; 2025).

These components form the building blocks of a general architecture for multi-scale forecasting.

**Definition 4.1** (Multi-scale forecasting model). *A multi-scale forecasting model for time series is a function that maps an input sequence $\boldsymbol{x}$ to a predicted output sequence $\hat{\boldsymbol{y}} \in \mathbb{R}^T$:*

$$\hat{\boldsymbol{y}} = \big( \bigcirc_{i=1}^{K} (p_{\theta_i} \circ g_i) \big)(\boldsymbol{x}|\boldsymbol{s}), \tag{3}$$

*where the operator $\bigcirc$ represents a composition of $K$ stacked blocks. The $i$-th block consists of a multi-scale generator $g_i$ and a multi-scale processor $p$ with parameters $\theta_i$.*

**Theorem 4.2.** *For each of the following multi-scale model architectures: (Liu et al., 2022b;a; Wu et al., 2023; Wang et al., 2023; Cai et al., 2024; Wang et al., 2024; 2025; Hu et al., 2025a; Murad et al., 2025; Naghashi et al., 2025), there exists a set of multi-scale generators $\{g_i\}_{i=1}^K$ and multi-scale processors $\{p_{\theta_i}\}_{i=1}^K$ such that the model is an instance of Definition 4.1.*

Definition 4.1 provides a unifying perspective that encompasses a wide range of existing methods in time series analysis. The depth $K$ specifies the number of generation-processing blocks, which determines the complexity of interactions across different scale parameters. A comprehensive categorization of existing models and the proof of Theorem 4.2 are provided in Appendix A.

## 4.2 LIMITATIONS OF EXISTING MULTI-SCALE METHODS

Based on our unifying framework of multi-scale methods, we first present two notable observations of these models fundamentally characterized by the design of their generators and processors.

**Observation 1. Static multi-scale generator** Existing models select a finite set of scale parameters $\boldsymbol{s} = \{s_1, \ldots, s_M\}$ in advance, e.g., window length, stride, and dilation, and consider them as fixed hyperparameters. A dyadic progression $s_i = 2^{i-1}$ is the most common choice due to its simplicity and computational efficiency (Wang et al., 2023; 2024; Hu et al., 2025a). Under fixed $\boldsymbol{s}$, the multi-scale generator applies an identical set of scale operators uniformly over the entire time series.

**Observation 2. Staged multi-scale processor** Multi-scale processors commonly adopt a two-stage approach. First, temporal dependencies are processed within each scaled representation in isolation. Second, a separate cross-scale fusion mechanism integrates the outputs to capture interactions across different levels of scaling. The fusion process follows a fixed traversal order, such as coarse-to-fine, fine-to-coarse, or stacked multi-pass schedules (Zhong et al., 2024; Wang et al., 2025).

From these observations, we identify two key limitations.

**Limitation 1. Rigid scaling strategy** Fixed sets of scales cannot adjust to each target dataset. Since time series exhibit diverse periodicities and trends, a pre-defined $\boldsymbol{s}$ is rarely optimal. Some scaling operator families reinforce this limitation: moving averages with fixed windows fail to capture long-range dependencies, while fixed-rate subsampling imposes grid patterns that overlook informative resolutions. Such designs limit the extensibility of multi-scale modeling.

**Limitation 2. Lack of flexibility** Staged multi-scale processors constrain both flexibility and efficiency. In the first stage, temporal dependencies and scale information are modeled in isolation, preventing intertwined relationships. In the second stage, cross-scale fusion depends on the completion of all per-scale computations, creating sequential bottlenecks. Explicitly modeling cross-scale dependencies further enforces sequential processing, hindering efficiency and parallelization.

## 5 SIGMA: SIMPLIFIED MULTI-SCALE MODELING

We introduce SIGMA, a simple yet principled framework for multi-scale time series modeling. Motivated by the limitations of existing approaches, SIGMA ensures learnable multi-scale generation and flexible processing for any given time series; it instantiates the multi-scale generator with the learnable discrete Gaussian (LDG) kernel grounded in scale-space theory, and adopts a lightweight MLP as the multi-scale processor. Detailed proofs for all theorems are provided in Appendix A.

## 5.1 SIMPLIFIED MULTI-SCALE GENERATOR

We first extend scaling operator families to have multiple, continuous scale parameters $s \in \mathbb{R}_+^M$ for each transformation $f$, unlike the single, discrete parameter in Definition 3.1.

**Definition 5.1** (Extended scaling operator family). *A set of transformations $\mathcal{F} = \{f(\boldsymbol{x}|\boldsymbol{s}) \mid \boldsymbol{s} \in \mathbb{R}_+^M\}$ with $f(\cdot|\boldsymbol{s}) : \mathcal{X} \to \mathcal{X}_{\boldsymbol{s}} \subset \mathbb{R}^{L_{\boldsymbol{s}}}$ is an extended scaling operator family if it satisfies*

- *Consistency: For $s \in \mathbb{Z}_+$, $\{f(\boldsymbol{x}|s\mathbf{1})\}$ is a scaling operator family.*
- *Differentiability: For any $\boldsymbol{x} \in \mathcal{X}$, $f(\boldsymbol{x}|\boldsymbol{s})$ is continuously differentiable with respect to $\boldsymbol{s}$.*

*The output dimension $L_{\boldsymbol{s}}$ depends on the set of scale parameters $\boldsymbol{s} = (s_1, \cdots, s_M)$.*

This generalized definition presents two key advantages. First, scale parameters $\boldsymbol{s}$ can be optimized directly from data via gradient-based methods, since $f(\boldsymbol{x}|\boldsymbol{s})$ is differentiable over $\mathbb{R}_+^M$. This allows the model to learn scale parameters from data rather than relying on fixed, manual choices. Second, a single extended operator $f$ can express a rich range of scales simultaneously, removing the need for stacked or hierarchical modules with discrete, pre-specified scale parameters. Together, these properties directly address the rigid scaling strategy identified in Section 4.2.

Grounded in these principles, a variety of constructions could realize an extended scaling operator family. In this work, we propose the *learnable discrete Gaussian* (LDG) kernel $k$, defined as

$$k(\boldsymbol{x}|\boldsymbol{s}) = \boldsymbol{K}(\boldsymbol{s})\boldsymbol{x}, \qquad [\boldsymbol{K}(\boldsymbol{s})]_{i,j} = e^{-s_i} I_{|i-j|}(s_i), \tag{4}$$

where $[\cdot]_{i,j}$ denotes the $(i,j)$-th element of a matrix, $\boldsymbol{K}(\boldsymbol{s}) \in \mathbb{R}^{L \times L}$ with $L$ being the length of the sequence $\boldsymbol{x}$, and $I_n(\cdot)$ is the modified Bessel function of the first kind of integer order $n$. We model the scale parameters $\boldsymbol{s}$ to be learnable. As a result, $k$ applies a dynamic scaling to each element of $\boldsymbol{x}$, allowing a rich transformation without manually deciding various scale parameters.

Figure 3 illustrates how the LDG kernel actually works; it yields smooth, scale-controlled transformations at each time step with learnable scale parameters. It also provides a symmetric and effectively unbounded receptive field for capturing long-range dependencies. Theorem 5.2 formally establishes that this kernel constitutes a valid extended scaling operator family, confirming its suitability as a principled generator. Moreover, this Gaussian instantiation is the uniquely determined operator by the discrete scale-space axioms as

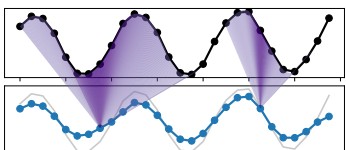

Figure 3: The LDG kernel performs smoothing with position-wise scales.

shown in Theorem 5.3. This result aligns our method with scale-space theory, a principled framework for multi-resolution signal analysis originating in computer vision (Witkin, 1987; Lindeberg, 2013).

**Theorem 5.2.** *The LDG kernel family, $\{k(\cdot|\boldsymbol{s}) \mid \boldsymbol{s} \in \mathbb{R}_+^M\}$, is an extended scaling operator family.*

**Theorem 5.3.** *The LDG kernel family is the unique extended scaling operator family of symmetric kernels that satisfies the discrete scale-space axioms (Lindeberg, 2002).*

## 5.2 SIMPLIFIED MULTI-SCALE PROCESSOR

To complement our principled generator, we propose a *simplified multi-scale processor* that overcomes the inflexibility of existing methods. In previous work, heuristic scaling produced disjoint representations, forcing complex processors to reconstruct cross-scale relationships. In contrast, our LDG kernel yields all scales from a single, continuous, theoretically grounded operator, preserving smooth inter-scale structure. As a result, instead of modeling temporal dependencies and cross-scale interactions in separate stages, we are able to process both simultaneously in a single step.

We implement this processor with a lightweight, two-layer multi-layer perceptron (MLP) operating on the concatenated multi-scale representations, followed by a linear projection to the forecasting horizon. By a single forward pass, it jointly refines temporal features and fuses information across scales, enabling efficient and flexible learning of nonlinear interactions without pre-wired pathways. This one-shot composition directly addresses the lack of flexibility identified in Section 4.2.

Given an input sequence $\boldsymbol{x} \in \mathbb{R}^L$, we first produce $d$-dimensional embeddings $\boldsymbol{X} = \text{Embed}(\boldsymbol{x}) \in \mathbb{R}^{L \times d}$ as done in (Wu et al., 2021; Wang et al., 2024; 2025). Based on these embeddings, SIGMA,

Table 2: Long-term forecasting results across eight datasets with horizons $T \in \{96, 192, 336, 720\}$ and input length fixed at 96. SIGMA achieves the smallest forecasting errors in 55 out of 80 evaluation settings and the second-best in 19 cases. These results confirm the effectiveness of principled multi-scale modeling for diverse long-horizon forecasting tasks.

| Method | | SIGMA (Ours) | | AMD (2025a) | | MultiPatch. (2025) | | WPMixer (2025) | | TimeMixer (2024) | | MSGNet (2024) | | MICN (2023) | | TimesNet (2023) | | Pyra. (2022b) | |
|---|---|---|---|---|---|---|---|---|---|---|---|---|---|---|---|---|---|---|---|
| Metric | | MSE | MAE | MSE | MAE | MSE | MAE | MSE | MAE | MSE | MAE | MSE | MAE | MSE | MAE | MSE | MAE | MSE | MAE |
| Weather | 96 | **0.160** | **0.204** | 0.182 | 0.227 | 0.172 | 0.211 | 0.164 | 0.210 | 0.166 | 0.214 | 0.161 | 0.209 | 0.192 | 0.250 | 0.172 | 0.221 | 0.195 | 0.281 |
| | 192 | 0.209 | **0.248** | 0.231 | 0.266 | 0.218 | 0.254 | 0.209 | 0.250 | **0.208** | 0.251 | 0.217 | 0.257 | 0.233 | 0.289 | 0.225 | 0.265 | 0.245 | 0.322 |
| | 336 | 0.270 | 0.293 | 0.283 | 0.302 | 0.275 | 0.296 | **0.264** | **0.290** | 0.265 | 0.293 | 0.280 | 0.303 | 0.283 | 0.332 | 0.289 | 0.309 | 0.307 | 0.365 |
| | 720 | 0.348 | 0.345 | 0.357 | 0.350 | 0.355 | 0.348 | **0.344** | **0.343** | 0.345 | 0.345 | 0.373 | 0.362 | 0.354 | 0.388 | 0.361 | 0.355 | 0.394 | 0.420 |
| | Avg | 0.247 | **0.273** | 0.263 | 0.286 | 0.255 | 0.278 | **0.245** | **0.273** | 0.246 | 0.276 | 0.258 | 0.283 | 0.266 | 0.315 | 0.262 | 0.288 | 0.285 | 0.347 |
| Electricity | 96 | **0.146** | **0.241** | 0.187 | 0.269 | 0.173 | 0.259 | 0.167 | 0.259 | 0.157 | 0.248 | 0.169 | 0.281 | 0.171 | 0.284 | 0.165 | 0.268 | 0.283 | 0.377 |
| | 192 | **0.163** | **0.257** | 0.191 | 0.274 | 0.181 | 0.267 | 0.179 | 0.268 | 0.169 | 0.260 | 0.189 | 0.298 | 0.178 | 0.290 | 0.182 | 0.284 | 0.296 | 0.391 |
| | 336 | **0.179** | **0.273** | 0.206 | 0.290 | 0.199 | 0.285 | 0.197 | 0.288 | 0.187 | 0.277 | 0.201 | 0.310 | 0.189 | 0.301 | 0.198 | 0.299 | 0.306 | 0.401 |
| | 720 | **0.213** | **0.304** | 0.248 | 0.323 | 0.239 | 0.318 | 0.232 | 0.315 | 0.227 | 0.312 | 0.238 | 0.340 | 0.208 | 0.318 | 0.231 | 0.325 | 0.305 | 0.393 |
| | Avg | **0.175** | **0.269** | 0.208 | 0.289 | 0.198 | 0.282 | 0.194 | 0.282 | 0.185 | 0.274 | 0.199 | 0.307 | 0.187 | 0.298 | 0.194 | 0.294 | 0.298 | 0.390 |
| Traffic | 96 | **0.431** | **0.288** | 0.544 | 0.345 | 0.471 | 0.318 | 0.528 | 0.347 | 0.477 | 0.309 | 0.599 | 0.353 | 0.516 | 0.309 | 0.590 | 0.318 | 0.678 | 0.384 |
| | 192 | **0.444** | **0.296** | 0.527 | 0.335 | 0.480 | 0.319 | 0.511 | 0.337 | 0.488 | 0.312 | 0.634 | 0.372 | 0.535 | 0.317 | 0.614 | 0.327 | 0.672 | 0.377 |
| | 336 | **0.461** | **0.303** | 0.538 | 0.339 | 0.499 | 0.329 | 0.519 | 0.337 | 0.506 | 0.319 | 0.663 | 0.391 | 0.548 | 0.322 | 0.640 | 0.342 | 0.681 | 0.381 |
| | 720 | **0.494** | **0.320** | 0.573 | 0.358 | 0.537 | 0.351 | 0.548 | 0.350 | 0.535 | 0.332 | 0.721 | 0.420 | 0.574 | 0.332 | 0.662 | 0.350 | 0.709 | 0.395 |
| | Avg | **0.458** | **0.302** | 0.546 | 0.344 | 0.497 | 0.329 | 0.527 | 0.343 | 0.501 | 0.318 | 0.654 | 0.384 | 0.543 | 0.320 | 0.627 | 0.334 | 0.685 | 0.384 |
| Exchange | 96 | 0.084 | 0.204 | **0.083** | **0.201** | 0.089 | 0.208 | 0.086 | 0.202 | 0.090 | 0.210 | 0.104 | 0.230 | 0.093 | 0.226 | 0.112 | 0.242 | 0.630 | 0.645 |
| | 192 | **0.174** | **0.297** | 0.175 | 0.297 | 0.187 | 0.308 | 0.176 | **0.296** | 0.185 | 0.305 | 0.200 | 0.322 | 0.184 | 0.331 | 0.214 | 0.334 | 0.935 | 0.782 |
| | 336 | **0.322** | **0.411** | 0.326 | 0.412 | 0.357 | 0.436 | 0.343 | 0.421 | 0.351 | 0.428 | 0.394 | 0.460 | 0.322 | 0.443 | 0.379 | 0.452 | 1.204 | 0.873 |
| | 720 | **0.833** | **0.687** | 0.847 | 0.693 | 0.913 | 0.724 | 0.900 | 0.712 | 0.911 | 0.713 | 1.027 | 0.772 | 0.780 | 0.694 | 0.961 | 0.746 | 1.956 | 1.117 |
| | Avg | **0.353** | **0.400** | 0.358 | 0.401 | 0.387 | 0.419 | 0.376 | 0.410 | 0.384 | 0.414 | 0.431 | 0.446 | 0.345 | 0.424 | 0.416 | 0.443 | 1.181 | 0.854 |
| ETTh1 | 96 | 0.379 | **0.393** | 0.385 | 0.396 | **0.377** | 0.397 | 0.382 | 0.404 | 0.381 | 0.398 | 0.398 | 0.418 | 0.425 | 0.435 | 0.419 | 0.432 | 0.701 | 0.630 |
| | 192 | 0.430 | **0.425** | 0.437 | **0.425** | 0.427 | 0.428 | 0.438 | 0.427 | 0.441 | 0.434 | 0.444 | 0.445 | 0.505 | 0.484 | 0.474 | 0.464 | 0.850 | 0.713 |
| | 336 | 0.481 | 0.446 | 0.480 | **0.445** | 0.469 | 0.449 | 0.499 | 0.464 | 0.475 | 0.449 | 0.484 | 0.471 | 0.606 | 0.556 | 0.499 | 0.475 | 0.960 | 0.777 |
| | 720 | **0.480** | **0.468** | 0.485 | 0.469 | 0.499 | 0.485 | 0.487 | 0.471 | 0.522 | 0.494 | 0.509 | 0.498 | 0.752 | 0.648 | 0.529 | 0.500 | 1.005 | 0.803 |
| | Avg | **0.443** | **0.433** | 0.447 | 0.434 | 0.443 | 0.440 | 0.451 | 0.442 | 0.455 | 0.444 | 0.459 | 0.458 | 0.572 | 0.531 | 0.480 | 0.468 | 0.879 | 0.731 |
| ETTh2 | 96 | **0.289** | **0.339** | 0.291 | 0.340 | 0.293 | 0.347 | 0.291 | 0.342 | 0.296 | 0.348 | 0.327 | 0.369 | 0.358 | 0.405 | 0.327 | 0.369 | 1.439 | 0.922 |
| | 192 | **0.369** | **0.394** | 0.373 | 0.391 | 0.372 | 0.396 | 0.376 | 0.397 | 0.375 | 0.395 | 0.408 | 0.417 | 0.497 | 0.484 | 0.404 | 0.412 | 5.640 | 1.894 |
| | 336 | **0.415** | **0.427** | 0.416 | 0.427 | 0.420 | 0.431 | 0.435 | 0.439 | 0.430 | 0.438 | 0.429 | 0.439 | 0.618 | 0.552 | 0.459 | 0.456 | 4.800 | 1.844 |
| | 720 | 0.431 | 0.446 | **0.424** | **0.441** | 0.431 | 0.450 | 0.459 | 0.462 | 0.456 | 0.460 | 0.446 | 0.459 | 0.856 | 0.666 | 0.451 | 0.459 | 4.466 | 1.824 |
| | Avg | **0.376** | 0.402 | 0.376 | **0.400** | 0.379 | 0.406 | 0.390 | 0.410 | 0.389 | 0.410 | 0.402 | 0.421 | 0.582 | 0.527 | 0.410 | 0.424 | 4.086 | 1.621 |
| ETTm1 | 96 | 0.323 | 0.359 | 0.330 | 0.365 | **0.319** | **0.358** | 0.322 | 0.358 | 0.322 | 0.359 | 0.328 | 0.370 | 0.322 | 0.373 | 0.334 | 0.374 | 0.592 | 0.514 |
| | 192 | **0.360** | **0.381** | 0.372 | 0.384 | 0.363 | 0.385 | 0.361 | 0.382 | 0.364 | 0.384 | 0.371 | 0.395 | 0.361 | 0.402 | 0.404 | 0.409 | 0.645 | 0.568 |
| | 336 | 0.392 | 0.404 | 0.406 | 0.405 | 0.398 | 0.410 | **0.389** | **0.402** | 0.397 | 0.407 | 0.410 | 0.419 | 0.409 | 0.437 | 0.418 | 0.421 | 0.776 | 0.643 |
| | 720 | 0.455 | **0.442** | 0.471 | **0.440** | 0.460 | 0.448 | 0.469 | 0.447 | 0.456 | 0.447 | 0.494 | 0.465 | 0.506 | 0.498 | 0.484 | 0.455 | 0.936 | 0.730 |
| | Avg | **0.383** | **0.397** | 0.395 | 0.399 | 0.385 | 0.400 | 0.385 | 0.397 | 0.385 | 0.398 | 0.401 | 0.412 | 0.399 | 0.427 | 0.410 | 0.415 | 0.737 | 0.614 |
| ETTm2 | 96 | **0.174** | **0.257** | 0.183 | 0.267 | 0.177 | 0.259 | 0.175 | 0.257 | 0.176 | 0.258 | 0.179 | 0.263 | 0.186 | 0.283 | 0.187 | 0.266 | 0.387 | 0.464 |
| | 192 | **0.239** | **0.299** | 0.246 | 0.306 | 0.243 | 0.304 | 0.240 | 0.299 | 0.241 | 0.302 | 0.250 | 0.308 | 0.278 | 0.352 | 0.257 | 0.309 | 0.676 | 0.623 |
| | 336 | **0.296** | **0.337** | 0.305 | 0.342 | 0.305 | 0.346 | 0.304 | 0.342 | 0.304 | 0.345 | 0.311 | 0.345 | 0.405 | 0.437 | 0.322 | 0.349 | 1.196 | 0.836 |
| | 720 | **0.394** | **0.394** | 0.404 | 0.397 | 0.407 | 0.404 | 0.398 | 0.397 | 0.405 | 0.402 | 0.416 | 0.406 | 0.546 | 0.515 | 0.427 | 0.409 | 3.588 | 1.460 |
| | Avg | **0.276** | **0.322** | 0.285 | 0.328 | 0.283 | 0.328 | 0.279 | 0.324 | 0.281 | 0.327 | 0.289 | 0.330 | 0.354 | 0.397 | 0.298 | 0.333 | 1.462 | 0.846 |

our multi-scale forecasting model, is succinctly represented by the following equation:

$$\hat{\boldsymbol{y}} = \boldsymbol{W}_1(\mathrm{MLP}(\boldsymbol{H}) + \boldsymbol{H})\boldsymbol{W}_2, \quad \boldsymbol{H} = \boldsymbol{K}(\boldsymbol{s})\boldsymbol{X} \, \| \, (\boldsymbol{I} - \boldsymbol{K}(\boldsymbol{s}))\boldsymbol{X} \in \mathbb{R}^{2L \times d}, \tag{5}$$

where $\boldsymbol{s} \in \mathbb{R}_+^L$ is the set of learnable scale parameters, and $\boldsymbol{W}_1 \in \mathbb{R}^{T \times 2L}$ and $\boldsymbol{W}_2 \in \mathbb{R}^{d \times 1}$ denote the projection heads. These scale parameters are optimized as dataset-level parameters during training and kept fixed at inference. We decompose the input into a smoothed component $\boldsymbol{K}(\boldsymbol{s})\boldsymbol{X}$ and a residual component $(\boldsymbol{I} - \boldsymbol{K}(\boldsymbol{s}))\boldsymbol{X}$ analogous to the classical trend-seasonal decomposition (Cleveland et al., 1990). To further stabilize optimization and preserve scale-specific information, we incorporate a skip connection that adds each scale's input back to its transformed output (He et al., 2016).

We adopt channel independence, processing each variable separately, which makes the architecture naturally applicable to multivariate settings (Zeng et al., 2023). We further apply reversible instance normalization to each time series variable to remove distribution shifts (Kim et al., 2022).

# 6 EXPERIMENTS

We conduct a comprehensive empirical evaluation of SIGMA on standard long-term and short-term forecasting datasets against eight state-of-the-art baseline models. We also provide deeper analyses to investigate its efficiency, robustness, and underlying design principles.

Table 3: Short-term forecasting results in the M4 benchmark dataset with various temporal granularities. SIGMA achieves the best performance in 11 of the 15 cases. This highlights its effectiveness in capturing multi-scale patterns across a diverse range of time series types.

| | Method | SIGMA (Ours) | AMD (2025a) | MultiPatch. (2025) | WPMixer (2025) | TimeMixer (2024) | MSGNet (2024) | MICN (2023) | TimesNet (2023) | Pyra. (2022b) |
|---|---|---|---|---|---|---|---|---|---|---|
| Yearly | SMAPE | 13.314 | 13.447 | **13.296** | 13.632 | 13.326 | 13.354 | 14.580 | 13.482 | 14.987 |
| | MASE | **2.989** | 3.022 | 3.009 | 3.075 | 3.002 | **2.989** | 3.382 | 3.056 | 3.361 |
| | OWA | **0.783** | 0.792 | 0.785 | 0.804 | 0.785 | 0.784 | 0.871 | 0.797 | 0.881 |
| Quarterly | SMAPE | **10.060** | 10.259 | 10.166 | 10.299 | 10.281 | 10.446 | 11.389 | 10.116 | 11.706 |
| | MASE | **1.177** | 1.211 | 1.178 | 1.217 | 1.206 | 1.248 | 1.380 | 1.186 | 1.397 |
| | OWA | **0.886** | 0.907 | 0.892 | 0.911 | 0.907 | 0.929 | 1.020 | 0.892 | 1.041 |
| Monthly | SMAPE | **12.750** | 12.898 | 12.810 | 12.945 | 12.984 | 12.970 | 13.797 | 12.775 | 14.444 |
| | MASE | **0.936** | 0.952 | 0.942 | 0.959 | 0.964 | 0.976 | 1.077 | 0.945 | 1.142 |
| | OWA | **0.882** | 0.895 | 0.887 | 0.900 | 0.903 | 0.908 | 0.985 | 0.887 | 1.038 |
| Others | SMAPE | 4.867 | 4.822 | 4.849 | 4.925 | **4.739** | 5.521 | 6.123 | 4.978 | 6.115 |
| | MASE | 3.316 | 3.245 | 3.271 | 3.259 | **3.234** | 3.829 | 4.196 | 3.265 | 4.156 |
| | OWA | 1.037 | 1.021 | 1.028 | 1.028 | **1.014** | 1.174 | 1.300 | 1.048 | 1.282 |
| Weighted Average | SMAPE | **11.840** | 11.987 | 11.889 | 12.067 | 12.002 | 12.080 | 13.015 | 11.910 | 13.495 |
| | MASE | **1.585** | 1.605 | 1.591 | 1.623 | 1.604 | 1.647 | 1.836 | 1.604 | 1.864 |
| | OWA | **0.868** | 0.881 | 0.872 | 0.887 | 0.882 | 0.898 | 0.983 | 0.876 | 1.015 |

**Datasets** We evaluate SIGMA on both long-term and short-term forecasting benchmarks, including (long-term) Weather, ETT (ETTh1, ETTh2, ETTm1, ETTm2), Electricity, Exchange-Rate, Traffic, and (short-term) M4. We follow the standard protocol and partition all datasets into training, validation, and test sets in chronological order, using a 6:2:2 ratio for the ETT datasets and a 7:1:2 ratio for the others (Zhou et al., 2021). Consistent with previous work, the observation window is fixed to $L = 96$, and forecasting horizons are set to $T \in \{96, 192, 336, 720\}$ for long-term forecasting tasks (Wu et al., 2023). The M4 benchmark consists of 100,000 univariate time series collected at multiple temporal frequencies ranging from yearly to hourly. We follow the official benchmark protocol, where the prediction lengths are fixed according to the frequency of each series (Makridakis et al., 2018). Comprehensive details for each dataset are provided in Appendix B.

**Baselines** We compare SIGMA against eight state-of-the-art baselines that adopt multi-scale modeling strategies: AMD (Hu et al., 2025a), MultiPatchFormer (Naghashi et al., 2025), WPMixer (Murad et al., 2025), TimeMixer (Wang et al., 2024), MSGNet (Cai et al., 2024), MICN (Wang et al., 2023), TimesNet (Wu et al., 2023) and Pyraformer (Liu et al., 2022b).

**Experimental settings** For each benchmark, we adopt evaluation metrics used in previous work. For long-term forecasting, we use mean squared error (MSE) and mean absolute error (MAE) (Wu et al., 2023). For the short-term benchmark, we report symmetric mean absolute percentage error (SMAPE), mean absolute scaled error (MASE), and the overall weighted average (OWA) (Oreshkin et al., 2020). We apply the same batch size, number of training epochs, and early-stopping strategy across all methods to maintain a unified setting (Wu et al., 2023). Each experiment is repeated three times, and the averaged results are reported. Additional details are provided in Appendix B.

## 6.1 COMPARISON WITH BASELINE MODELS

**Long-term forecasting** Table 2 shows that SIGMA achieves the best performance in 55 out of 80 evaluation settings for long-term forecasting across all datasets and horizons. The improvements are particularly seen in the high-dimensional Electricity and Traffic datasets. SIGMA achieves 5.4% and 7.8% reduction in MSE on average relative to the second-best model, respectively. It highlights the importance of learning position-wise scale parameters from data to capture fine-grained temporal dynamics through principled multi-scale modeling when applied to real-world multivariate time series. In comparison to the competitors that employ complex multi-scale architectures, the performance improvement of SIGMA with a simple, parameter-efficient architecture further verifies the significance of our design.

**Short-term forecasting** Table 3 shows that SIGMA also establishes clear superiority over the competitors on the M4 benchmark for short-term forecasting, achieving the best results in 11 out of 15 cases. It achieves the best average results, particularly in the Quarterly and Monthly datasets, which contain the largest number of series. On the other hand, SIGMA exhibits limited performance on the "Others" category, which lacks sufficient data as it accounts for less than 5% of the benchmark. This

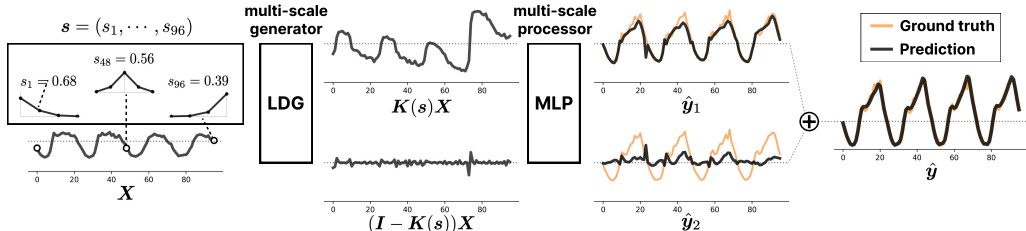

Figure 5: Case study on Traffic for predict-96 setting. The multi-scale generator employs the LDG kernel to extract trend-seasonal representations of the time series, while the multi-scale processor uses an MLP to produce complementary coarse- and fine-grained components. By integrating these complementary signals, SIGMA achieves more accurate and effective predictions.

suggests that learning position-wise scale parameters is most effective when trained on sufficiently large datasets, while less-structured series favor stronger inductive biases to extract meaningful patterns. Nevertheless, SIGMA outperforms all other baselines by a large margin overall.

## 6.2 DEEPER ANALYSIS ON SIGMA

**Ablation study**  To validate the design principles of SIGMA, we conduct an ablation study on the ETTh1 dataset under the predict-720 setting, comparing SIGMA against five of its variants: ① integrates the trend-seasonal mixing multi-scale processor from TimeMixer with LDG kernel multi-scale generator; ② changes the LDG kernel to have a single scale parameter applied to all elements; ③ removes the scaling mechanism entirely and relies only on the raw input; ④ employs moving average, a non-learnable scale operator family; ⑤ uses unnormalized convolution, a learnable kernel that does not belong to the scale operator family.

Table 4: Ablation study on ETTh1 with predict-720 setting.

| Method | MSE | MAE |
|--------|-----|-----|
| SIGMA | **0.480** | 0.468 |
| ① | 0.486 | **0.467** |
| ② | 0.489 | 0.473 |
| ③ | 0.492 | 0.475 |
| ④ | 0.493 | 0.475 |
| ⑤ | 0.524 | 0.492 |

As shown in Table 4, most modifications result in performance degradation. This indicates that the expressivity of the LDG kernel in ① and ②, the effect of scaling in ③, and the position-wise learnable scaling in ④ all play a critical role. While ① achieves performance comparable to SIGMA through a different multi-scale processor design, its multi-stage mixing architecture introduces additional computational overhead, whereas similar accuracy can be achieved with a simpler MLP-based processor. The significant performance drop in ⑤ demonstrates that invalid scaling operations which do not belong to the scale operator family can be sub-optimal, as they can transform the inherent characteristic of time series as well.

**Efficiency analysis**  We assess efficiency in terms of per-iteration (i.e., batch) training time and memory footprint on the ETTh1 dataset under the predict-720 setting, with all methods executed on a single GPU using the same batch size. As shown in Figure 4, SIGMA attains the fastest training time and the most compact memory footprint while achieving the lowest MSE. Specifically, SIGMA reduces memory consumption by 3.8 times and improves training speed by 5.3 times compared to AMD, the previous state-of-the-art baseline.

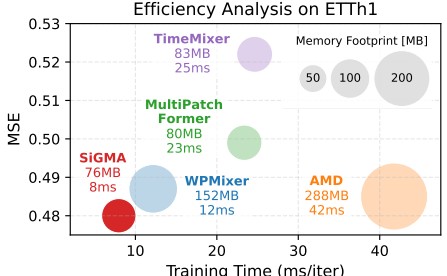

Figure 4: Efficiency analysis on ETTh1 with predict-720 setting. SIGMA achieves the best trade-off between accuracy and efficiency.

This significant efficiency gap comes from fundamental architectural differences. AMD stacks three downsampling stages and applies per-variable multi-scale mixing in a sequential manner. While this approach may improve accuracy over other baselines, it imposes substantial memory and runtime overhead. In contrast, SIGMA employs a single-kernel formulation that preserves full parallelism in its representation updates. Our principled simplification of multi-scale modeling yields a more favorable speed-memory trade-off without compromising modeling capacity, leading to superior overall performance.

**Case study** We conduct a case study on the Traffic dataset under the predict-96 setting to illustrate the behavior of SIGMA. As shown in Figure 5, SIGMA learns a scale parameter at each timestep. The LDG kernel acts as a principled smoothing operator, allowing the model to selectively emphasize broad trends or localized variations. The resulting representations are then processed by the MLP, which captures richer temporal dependencies and adjusts the balance between coarse- and fine-grained components. By integrating these components, SIGMA yields predictions that closely align with the ground truth, validating the effectiveness of its multi-scale design.

## 7 CONCLUSION

In this work, we introduce a concept of *scaling operator families,* which allows us to unify existing multi-scale models within a principled framework and decompose them into a pair of a multi-scale generator and a multi-scale processor. Building on this foundation, we propose SIGMA, a simple yet principled architecture that employs the learnable discrete Gaussian (LDG) kernel to enable expressive scaling, where a lightweight MLP processor is then applied for efficient cross-scale fusion. Extensive experiments on both long- and short-term forecasting benchmarks demonstrate that SIGMA consistently outperforms state-of-the-art baselines, while substantially reducing memory consumption and computational time. These findings underscore the effectiveness of SIGMA as a reliable and generalizable solution for time-series forecasting. Future work can include extending our framework to multivariate scaling and sample-specific dynamic modeling to further enhance its expressiveness and applicability to cover a more diverse range of time series data.

## REPRODUCIBILITY STATEMENT

Code and scripts to reproduce all experiments are available at `https://anonymous.4open.science/r/SiGMA-ICLR2026` and also included in the supplementary material. We ensure reproducibility by using publicly available datasets with standard splits and protocols. All experiments are conducted in PyTorch under unified training and evaluation settings across baselines. We report results averaged over three runs, and provide ablation studies and efficiency analyses for verification.

## ETHICS STATEMENT

This work develops a general framework for time-series forecasting and does not involve sensitive personal data. The potential impacts include positive applications such as improving forecasting in energy, climate, and transportation. We encourage responsible use of our method in domains that align with ethical and societal values.

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

# A PROOFS

## A.1 PROOF OF THEOREM 3.2

**Lemma A.1.** *If $T : \mathbb{R}^m \to \mathbb{R}^n$ satisfies $\|T(\boldsymbol{x}) - T(\boldsymbol{y})\|_2 \le \|\boldsymbol{x} - \boldsymbol{y}\|_2$ for all $\boldsymbol{x}, \boldsymbol{y}$, then $\|T(\boldsymbol{x})\|_2 \le \|\boldsymbol{x}\|_2$ for all $\boldsymbol{x}$.*

*Proof.* $\|T(\boldsymbol{x})\|_2 = \|T(\boldsymbol{x}) - T(\mathbf{0})\|_2 \le \|\boldsymbol{x} - \mathbf{0}\|_2 = \|\boldsymbol{x}\|_2.$ □

**Lemma A.2.** *Suppose for scales $s_i, s_j$ we have $f(\cdot|s_i) = G_{i,j} \circ f(\cdot|s_j)$ for some non-expansive $G_{i,j}$. Then, $\|f(\boldsymbol{x}|s_i)\|_2 \le \|f(\boldsymbol{x}|s_j)\|_2$. for all $\boldsymbol{x}$. Moreover, if there exists some $\boldsymbol{y}$ in the range of $f(\cdot|s_j)$ with $\|G_{i,j}(\boldsymbol{y})\|_2 < \|\boldsymbol{y}\|_2$, then there exists $\boldsymbol{x}' \in \mathcal{X}$ such that $\|f(\boldsymbol{x}'|s_i)\|_2 < \|f(\boldsymbol{x}'|s_j)\|_2$.*

*Proof.* By Lemma A.1, we have $\|G_{i,j}(\boldsymbol{u})\|_2 \le \|\boldsymbol{u}\|_2$ for all $\boldsymbol{u}$. Thus for any $\boldsymbol{x}$,

$$\|f(\boldsymbol{x}|s_i)\|_2 = \|G_{i,j}(f(\boldsymbol{x}|s_j))\|_2 \le \|f(\boldsymbol{x}|s_j)\|_2.$$

If some $\boldsymbol{y}$ in the range of $f(\cdot|s_j)$ satisfies $\|G_{i,j}(\boldsymbol{y})\|_2 < \|\boldsymbol{y}\|_2$, pick $\boldsymbol{x}'$ with $f(\boldsymbol{x}'|s_j) = \boldsymbol{y}$. Then

$$\|f(\boldsymbol{x}'|s_i)\|_2 = \|G_{i,j}(\boldsymbol{y})\|_2 < \|\boldsymbol{y}\|_2 = \|f(\boldsymbol{x}'|s_j)\|_2.$$

□

**Average pooling.** Define

$$\big(f_{\mathrm{avg}}(\boldsymbol{x}|s)\big)_m := \frac{1}{s} \sum_{i \in B_m} x_i,$$

where $B_m = \{(m-1)s+1, \ldots, ms\}$ for $m = 1, \ldots, L_s := P/s$.

*Non-expansiveness.* For any $\boldsymbol{x}, \boldsymbol{y}$ and any block $B_m$,

$$\big|f_{\mathrm{avg}}(\boldsymbol{x}|s)_m - f_{\mathrm{avg}}(\boldsymbol{y}|s)_m\big| = \frac{1}{s}\Big| \sum_{i \in B_m} (x_i - y_i)\Big|$$

$$\le \frac{1}{s}\sqrt{s}\, \|\boldsymbol{x}|_{B_m} - \boldsymbol{y}|_{B_m}\|_2 \le \|\boldsymbol{x}|_{B_m} - \boldsymbol{y}|_{B_m}\|_2.$$

Summing squares over $m$ yields

$$\|f_{\mathrm{avg}}(\boldsymbol{x}|s) - f_{\mathrm{avg}}(\boldsymbol{y}|s)\|_2 \le \|\boldsymbol{x} - \boldsymbol{y}\|_2,$$

so non-expansiveness holds.

*Energy reduction.* Let $s_i = m \cdot s_j$ with $m > 1$. Each $s_i$-block is a union of $m$ consecutive $s_j$-blocks. Let $y = f_{\mathrm{avg}}(\boldsymbol{x}|s_j) \in \mathbb{R}^{L_{s_j}}$, and define

$$\big(G_{i,j}(y)\big)_k := \frac{1}{m} \sum_{r=0}^{m-1} y_{km+r},$$

i.e., average pooling on $y$ with block size $m$. Then one checks directly that

$$f_{\mathrm{avg}}(\boldsymbol{x}|s_i) = G_{i,j}\big(f_{\mathrm{avg}}(\boldsymbol{x}|s_j)\big),$$

and $G_{i,j}$ is the same blockwise averaging operator as above, hence non-expansive. By Lemma A.2, for all $\boldsymbol{x}$,

$$\|f_{\mathrm{avg}}(\boldsymbol{x}|s_i)\|_2 \le \|f_{\mathrm{avg}}(\boldsymbol{x}|s_j)\|_2.$$

To see strictness for some input, choose $\boldsymbol{y}$ with two distinct values in each $m$-block, so that the block average has strictly smaller energy than $\boldsymbol{y}$. Since $f_{\mathrm{avg}}(\cdot|s_j)$ is surjective onto $\mathbb{R}^{L_{s_j}}$, there exists $\boldsymbol{x}'$ with $f_{\mathrm{avg}}(\boldsymbol{x}'|s_j) = \boldsymbol{y}$, and Lemma A.2 yields

$$\|f_{\mathrm{avg}}(\boldsymbol{x}'|s_i)\|_2 < \|f_{\mathrm{avg}}(\boldsymbol{x}'|s_j)\|_2.$$

**Max- and min-pooling.** Define
$$\left(f_{\max}(\boldsymbol{x}|s)\right)_m := \max_{i \in B_m} x_i, \; \left(f_{\min}(\boldsymbol{x}|s)\right)_m := \min_{i \in B_m} x_i.$$

*Non-expansiveness.* For any $\boldsymbol{u}, \boldsymbol{v} \in \mathbb{R}^s$,
$$\left| \max_i u_i - \max_i v_i \right| \le \max_i |u_i - v_i| \le \|\boldsymbol{u} - \boldsymbol{v}\|_2,$$
and similarly for the minimum. Thus each block map is 1-Lipschitz; by summing over blocks we get
$$\|f_{\max}(\boldsymbol{x}|s) - f_{\max}(\boldsymbol{y}|s)\|_2 \le \|\boldsymbol{x} - \boldsymbol{y}\|_2$$
and likewise for $f_{\min}$.

*Energy reduction.* Let $s_i = m \cdot s_j$ with $m > 1$. Define $y = f_{\max}(\boldsymbol{x}|s_j)$ and
$$\left(G_{i,j}(y)\right)_k := \max_{r \in \{0, \cdots, m-1\}} y_{km+r},$$
i.e., max-pooling over $m$ consecutive entries of $y$. Then
$$f_{\max}(\boldsymbol{x}|s_i) = G_{i,j}\left(f_{\max}(\boldsymbol{x}|s_j)\right),$$
and the same max-Lipschitz argument as above shows $G_{i,j}$ is non-expansive. Thus by Lemma A.2, for all $\boldsymbol{x}$,
$$\|f_{\max}(\boldsymbol{x}|s_i)\|_2 \le \|f_{\max}(\boldsymbol{x}|s_j)\|_2.$$

To get strictness, choose $\boldsymbol{y}$ such that in each group of $m$ entries there is at least one strictly smaller than the maximum; then $\|G_{i,j}(\boldsymbol{y})\|_2 < \|\boldsymbol{y}\|_2$. As before, we can realize such a $\boldsymbol{y}$ as $f_{\max}(\boldsymbol{x}'|s_j)$ by setting one large value and smaller ones within each block. The same reasoning applies to min-pooling by symmetry.

**Moving average.** Define
$$\left(f_{\mathrm{ma}}(\boldsymbol{x}|s)\right)_t := \frac{1}{s} \sum_{i=0}^{s-1} x_{t-i}, \qquad t = 1, \ldots, P.$$
This is convolution with $h_s = \frac{1}{s}\mathbf{1}_s$.

*Non-expansiveness.* By Young's inequality,
$$\|f_{\mathrm{ma}}(\boldsymbol{x}|s) - f_{\mathrm{ma}}(\boldsymbol{y}|s)\|_2 = \|h_s * (\boldsymbol{x} - \boldsymbol{y})\|_2 \le \|h_s\|_1 \|\boldsymbol{x} - \boldsymbol{y}\|_2 = \|\boldsymbol{x} - \boldsymbol{y}\|_2.$$
Thus $f_{\mathrm{ma}}(\cdot|s)$ is non-expansive.

*Energy reduction.* Let $X(\omega)$ and $Y_s(\omega)$ denote the Fourier transforms of $\boldsymbol{x}$ and $f_{\mathrm{ma}}(\boldsymbol{x}|s)$, respectively, and $H_s(\omega)$ the frequency response of $h_s$. We have
$$Y_s(\omega) = H_s(\omega)X(\omega), \; H_s(\omega) = \frac{1}{s}\sum_{k=0}^{s-1} e^{-i\omega k} = \frac{1}{s}\frac{\sin(s\omega/2)}{\sin(\omega/2)}e^{-i\omega(s-1)/2}.$$
Thus
$$\|f_{\mathrm{ma}}(\boldsymbol{x}|s)\|_2^2 = \frac{1}{2\pi}\int_{-\pi}^{\pi} |H_s(\omega)|^2 |X(\omega)|^2 \, d\omega.$$

For $s_i = m \cdot s_j$, write $\alpha = s_j\omega/2$. Then
$$\frac{|H_{s_i}(\omega)|}{|H_{s_j}(\omega)|} = \frac{1}{m}\frac{|\sin(m\alpha)|}{|\sin(\alpha)|} \le 1,$$
since $|\sin(m\alpha)| \le m|\sin(\alpha)|$ for all $m \in \mathbb{N}$ and all $\alpha$. Hence $|H_{s_i}(\omega)|^2 \le |H_{s_j}(\omega)|^2$ for all $\omega$, and therefore, for any $\boldsymbol{x}$,
$$\|f_{\mathrm{ma}}(\boldsymbol{x}|s_i)\|_2^2 \le \|f_{\mathrm{ma}}(\boldsymbol{x}|s_j)\|_2^2.$$

To see when the inequality is strict, define

$$D(\omega) := |H_{s_j}(\omega)|^2 - |H_{s_i}(\omega)|^2 \geq 0.$$

The above argument shows $D(\omega) \geq 0$ for all $\omega$, and $H_{s_i} \not\equiv H_{s_j}$ implies that $D(\omega) > 0$ on a set of nonzero measure. For any $\boldsymbol{x}$,

$$\|f_{\mathrm{ma}}(\boldsymbol{x}|s_j)\|_2^2 - \|f_{\mathrm{ma}}(\boldsymbol{x}|s_i)\|_2^2 = \frac{1}{2\pi} \int_{-\pi}^{\pi} D(\omega) \, |X(\omega)|^2 \, d\omega.$$

Therefore,

$$\|f_{\mathrm{ma}}(\boldsymbol{x}|s_j)\|_2^2 > \|f_{\mathrm{ma}}(\boldsymbol{x}|s_i)\|_2^2$$

for every input $\boldsymbol{x}$ whose spectrum $|X(\omega)|^2$ places nonzero mass on the region where $D(\omega) > 0$. Equivalently, the inequality is strict for any non-degenerate $\boldsymbol{x}$ whose energy is not concentrated entirely on the set where $|H_{s_i}(\omega)| = |H_{s_j}(\omega)|$.

**Subsampling.** Define

$$\big(f_{\mathrm{sub}}(\boldsymbol{x}|s)\big)_m := x_{(m-1)s+1}, \qquad L_s = P/s.$$

*Non-expansiveness.* For each block $B_m$, define $g_s(\mathbf{u}) = u_1$. Then

$$|g_s(\mathbf{u}) - g_s(\mathbf{v})| = |u_1 - v_1| \leq \|\mathbf{u} - \mathbf{v}\|_2.$$

Applying the blockwise arguments shows that $f_{\mathrm{sub}}(\cdot|s)$ is non-expansive.

*Energy reduction.* If $s_i = m \cdot s_j$, pure decimation satisfies

$$f_{\mathrm{sub}}(\boldsymbol{x}|s_i) = f_{\mathrm{sub}}\big(f_{\mathrm{sub}}(\boldsymbol{x}|s_j)\big|m\big).$$

So $f_{\mathrm{sub}}(\cdot|s_i) = G_{i,j} \circ f_{\mathrm{sub}}(\cdot|s_j)$ with $G_{i,j}$ another subsampling operator, which is non-expansive. Hence Lemma A.2 yields

$$\|f_{\mathrm{sub}}(\boldsymbol{x}|s_i)\|_2 \leq \|f_{\mathrm{sub}}(\boldsymbol{x}|s_j)\|_2$$

for all $\boldsymbol{x}$. The strict inequality holds by setting $\boldsymbol{x}$ as nonzero values concentrated on indices that are dropped at the coarser stride.

**Segmentation.** Define

$$f_{\mathrm{seg}}(\boldsymbol{x}|s) := (x_1, \ldots, x_{L_s}) \in \mathbb{R}^{L_s}, \qquad L_s = P/s,$$

*Non-expansiveness.* This is an orthogonal projection onto the first $L_s$ coordinates:

$$f_{\mathrm{seg}}(\boldsymbol{x}|s) = R_s \boldsymbol{x}, \quad R_s R_s^\top = I_{L_s}.$$

Thus

$$\|f_{\mathrm{seg}}(\boldsymbol{x}|s) - f_{\mathrm{seg}}(\boldsymbol{y}|s)\|_2 = \|R_s(\boldsymbol{x} - \boldsymbol{y})\|_2 \leq \|\boldsymbol{x} - \boldsymbol{y}\|_2,$$

so $f_{\mathrm{seg}}$ is non-expansive and $\|f_{\mathrm{seg}}(\boldsymbol{x}|s)\|_2 \leq \|\boldsymbol{x}\|_2$.

*Energy reduction.* If $s_i = m \cdot s_j$, then

$$f_{\mathrm{seg}}(\boldsymbol{x}|s_i) = R_{s_i} \boldsymbol{x} = R_{s_i} R_{s_j}^\top f_{\mathrm{seg}}(\boldsymbol{x}|s_j).$$

Here $G_{i,j} := R_{s_i} R_{s_j}^\top$ is an orthogonal projection from $\mathbb{R}^{L_{s_j}}$ to $\mathbb{R}^{L_{s_i}}$, hence non-expansive. Thus by Lemma A.2, for all $\boldsymbol{x}$,

$$\|f_{\mathrm{seg}}(\boldsymbol{x}|s_i)\|_2 \leq \|f_{\mathrm{seg}}(\boldsymbol{x}|s_j)\|_2.$$

The inequality is strict exactly when at least one of the discarded coordinates is nonzero, i.e., when

$$x_k \neq 0 \quad \text{for some } k \in \{L_{s_i} + 1, \ldots, L_{s_j}\}.$$

For such $\boldsymbol{x}$ we have

$$\sum_{k=L_{s_i}+1}^{L_{s_j}} x_k^2 > 0 \Rightarrow \|f_{\mathrm{seg}}(\boldsymbol{x}|s_i)\|_2 < \|f_{\mathrm{seg}}(\boldsymbol{x}|s_j)\|_2.$$

**Wavelet decomposition.** Let $\{\phi_{s,k}\}_k$ be an orthonormal basis of $V_s$, and define the wavelet approximation coefficients by

$$f_{\text{wav}}(\boldsymbol{x}|s)_k := \langle \boldsymbol{x}, \phi_{s,k} \rangle.$$

For standard wavelets, the approximation spaces $\{V_s\}$ are typically constructed as nested subspaces indexed by dyadic scales $s = 2^j$. Here, we consider a more general multiplicative scale set $\mathcal{S} \subset \mathbb{Z}_+$ and assume that to each $s \in \mathcal{S}$ we associate an approximation subspace $V_s \subset \mathbb{R}^{L_s}$ with orthogonal projector $P_s : \mathbb{R}^L \to V_s$, such that

$$s_i = ms_j \text{ with } m > 1 \quad \implies \quad V_{s_i} \subset V_{s_j} \text{ and } P_{s_i} = P_{s_i}P_{s_j}.$$

Equivalently, if $C_s : V_s \to \mathbb{R}^{L_s}$ denotes the isometry mapping $P_s\boldsymbol{x}$ to its coordinates in the basis $\{\phi_{s,k}\}_k$, then $f_{\text{wav}}(\boldsymbol{x}|s) = C_s P_s \boldsymbol{x} \in \mathbb{R}^{L_s}$.

*Non-expansiveness.* For any $s \in \mathcal{S}$, $P_s$ is an orthogonal projector, hence

$$\|P_s\boldsymbol{x} - P_s\boldsymbol{y}\|_2 \le \|\boldsymbol{x} - \boldsymbol{y}\|_2.$$

Since $C_s$ is an isometry on $V_s$,

$$\|f_{\text{wav}}(\boldsymbol{x}|s) - f_{\text{wav}}(\boldsymbol{y}|s)\|_2 = \|C_s(P_s\boldsymbol{x} - P_s\boldsymbol{y})\|_2 = \|P_s\boldsymbol{x} - P_s\boldsymbol{y}\|_2 \le \|\boldsymbol{x} - \boldsymbol{y}\|_2,$$

so $f_{\text{wav}}(\cdot|s)$ is non-expansive.

*Energy reduction.* Fix $s_i, s_j \in \mathcal{S}$ with $s_i = ms_j, m > 1$. For any $\boldsymbol{x}$, we can write

$$f_{\text{wav}}(\boldsymbol{x}|s_j) = C_{s_j} P_{s_j} \boldsymbol{x}, \quad f_{\text{wav}}(\boldsymbol{x}|s_i) = C_{s_i} P_{s_i} \boldsymbol{x} = C_{s_i} P_{s_i} P_{s_j} \boldsymbol{x}.$$

Define a linear map $G_{i,j} : \mathbb{R}^{L_{s_j}} \to \mathbb{R}^{L_{s_i}}$ by

$$G_{i,j} := C_{s_i} P_{s_i} P_{s_j}^\top C_{s_j}^\top, \quad \text{so that} \quad f_{\text{wav}}(\boldsymbol{x}|s_i) = G_{i,j}\big(f_{\text{wav}}(\boldsymbol{x}|s_j)\big).$$

Here $C_{s_j}, C_{s_i}$ are isometries and $P_{s_i}$ is an orthogonal projector, so

$$\|G_{i,j}\|_2 \le 1,$$

i.e., $G_{i,j}$ is non-expansive. By Lemma A.2, for all $\boldsymbol{x}$,

$$\|f_{\text{wav}}(\boldsymbol{x}|s_i)\|_2 = \|G_{i,j}(f_{\text{wav}}(\boldsymbol{x}|s_j))\|_2 \le \|f_{\text{wav}}(\boldsymbol{x}|s_j)\|_2.$$

For strictness, pick any $\boldsymbol{x} \in V_{s_j} \setminus V_{s_i}$. Then $P_{s_j}\boldsymbol{x} = \boldsymbol{x}$, while $P_{s_i}\boldsymbol{x}$ is the orthogonal projection of $\boldsymbol{x}$ onto the strictly smaller subspace $V_{s_i}$, so by Pythagoras theorem, $\|P_{s_i}\boldsymbol{x}\|_2 < \|\boldsymbol{x}\|_2$. Then,

$$\|f_{\text{wav}}(\boldsymbol{x}|s_i)\|_2 = \|C_{s_i} P_{s_i} \boldsymbol{x}\|_2 < \|C_{s_j}\boldsymbol{x}\|_2 = \|f_{\text{wav}}(\boldsymbol{x}|s_j)\|_2.$$

## A.2 PROOF OF THEOREM 3.3

*Proof.* We will show that each of operations is scale-degenerate in the sense that, for some pair $s_i > s_j$, we have

$$\|f(\boldsymbol{x}|s_i)\|_2 = \|f(\boldsymbol{x}|s_j)\|_2 \quad \text{for all } \boldsymbol{x} \in \mathcal{X},$$

so there exists no input $\boldsymbol{x}$ that yields strict inequality. In other words, all such families are degenerate in the scale parameter and therefore violate the energy reduction condition in Definition 3.1.

**Constant mappings.** If $f(\boldsymbol{x}|s) = \boldsymbol{c}$ for some fixed $\boldsymbol{c} \in \mathbb{R}^{L_s}$, then

$$\|f(\boldsymbol{x}|s_i)\|_2 = \|f(\boldsymbol{x}|s_j)\|_2 = \|\boldsymbol{c}\|_2.$$

**Permutations.** If $f(\boldsymbol{x}|s) = \Pi_s \boldsymbol{x}$ for a permutation matrix $\Pi_s$, then

$$\|f(\boldsymbol{x}|s_i)\|_2 = \|f(\boldsymbol{x}|s_j)\|_2 = \|\boldsymbol{x}\|_2.$$

**Additive shifts.** If $f(\boldsymbol{x}|s) = \boldsymbol{x} + \boldsymbol{b}$, then

$$\|f(\boldsymbol{x}|s_i)\|_2 = \|f(\boldsymbol{x}|s_j)\|_2 = \|\boldsymbol{x} + \boldsymbol{b}\|_2.$$

**Scalar multiplications.** If $f(\boldsymbol{x}|s) = c\boldsymbol{x}$ for a constant $c \in \mathbb{R}$, then

$$\|f(\boldsymbol{x}|s_i)\|_2 = \|f(\boldsymbol{x}|s_j)\|_2 = |c|\|\boldsymbol{x}\|_2.$$

**General linear maps.** If $f(\boldsymbol{x}|s) = \boldsymbol{W}\boldsymbol{x}$, then

$$\|f(\boldsymbol{x}|s_i)\|_2 = \|f(\boldsymbol{x}|s_j)\|_2 = \|\boldsymbol{W}\boldsymbol{x}\|_2.$$

$\square$

### A.3 PROOF OF THEOREM 4.2

*Proof.* Each listed architecture admits a decomposition into (i) multi-scale generators $g_i$ and (ii) multi-scale processors $p_{\theta_i}$. We specify $(K, \{g_i\}, \{p_{\theta_i}\})$ for each.

Here is a cleaner, camera-ready rewrite that keeps all your formulas intact, standardizes notation to $\boldsymbol{x}$, and tightens phrasing/formatting.

**Pyraformer (Liu et al., 2022b).**
- **Choice of $K$.** $K = 1$.
- **Generator $g_1$ (CSCM: strided-conv subsampling and pyramid).** Let $\boldsymbol{x}^{(0)} = \boldsymbol{x} \in \mathbb{R}^{L \times d}$ and fix $C \geq 2$ as a stride and size of kernel. For $s = 1, \ldots, M$ define

$$\boldsymbol{x}^{(s)} = \text{Conv1D}(\boldsymbol{x}^{(s-1)}; \text{kernel} = C, \text{stride} = C) \in \mathbb{R}^{L_s \times d_s}, \qquad L_s = \left\lfloor \frac{L}{C^s} \right\rfloor.$$

- **Processor $p_{\theta_1}$ (PAM).** Index pyramid nodes by $u = (s, t)$ and let $\mathcal{N}(u)$ be the intra-, inter-scale neighborhood. With multi-head projections

$$Q_u = \boldsymbol{x}_u W_Q^{(s)}, \quad K_v = \boldsymbol{x}_v W_K^{(s(v))}, \quad V_v = \boldsymbol{x}_v W_V^{(s(v))},$$

define

$$\alpha_{uv} = \frac{\exp(\langle Q_u, K_v \rangle / \sqrt{d_k})}{\sum_{w \in \mathcal{N}(u)} \exp(\langle Q_u, K_w \rangle / \sqrt{d_k})}, \qquad Z_u = \sum_{v \in \mathcal{N}(u)} \alpha_{uv} V_v,$$

and residual connection, feed-forward network update

$$Y_u = \text{LN}\big(\text{LN}(\boldsymbol{x}_u + Z_u) + \text{FFN}(\text{LN}(\boldsymbol{x}_u + Z_u))\big).$$

Upsample each $Y^{(s)}$ to length $L$ via $U_s$ and fuse:

$$H = \text{Fuse}\big(U_0 Y^{(0)}, \ldots, U_M Y^{(M)}\big) \in \mathbb{R}^{L \times d}, \qquad \hat{\boldsymbol{y}} = \text{TP}(H) \in \mathbb{R}^T.$$

**SCINet (Liu et al., 2022a).**
- **Choice of $K$.** $K = L$, which is the number of SCI levels.
- **Generator $g_i$ (even/odd subsampling).** Let $\boldsymbol{x} \in \mathbb{R}^{L \times d}$ be the current 1D sequence at level $i$. Define

$$S_{\text{even}}(\boldsymbol{x}) = (\boldsymbol{x}_0, \boldsymbol{x}_2, \ldots) \in \mathbb{R}^{\lceil L/2 \rceil \times d}, \quad S_{\text{odd}}(\boldsymbol{x}) = (\boldsymbol{x}_1, \boldsymbol{x}_3, \ldots) \in \mathbb{R}^{\lfloor L/2 \rfloor \times d},$$

and set

$$g_i(\boldsymbol{x}) = (\boldsymbol{x}_{\text{even}}, \boldsymbol{x}_{\text{odd}}), \qquad \boldsymbol{x}_{\text{even}} = S_{\text{even}}(\boldsymbol{x}), \; \boldsymbol{x}_{\text{odd}} = S_{\text{odd}}(\boldsymbol{x}).$$

- **Processor $p_{\theta_i}$ (SCI-Block: cross-gating, interaction, re-interleaving).** Let $\phi, \psi, \rho, \eta : \mathbb{R}^{* \times d} \to \mathbb{R}^{* \times d}$ be 1D conv modules. Cross-gating:

$$\boldsymbol{x}'_{\text{odd}} = \boldsymbol{x}_{\text{odd}} \odot \exp\big(\phi(\boldsymbol{x}_{\text{even}})\big), \qquad \boldsymbol{x}'_{\text{even}} = \boldsymbol{x}_{\text{even}} \odot \exp\big(\psi(\boldsymbol{x}_{\text{odd}})\big).$$
$$\tilde{\boldsymbol{x}}_{\text{odd}} = \boldsymbol{x}'_{\text{odd}} \pm \rho(\boldsymbol{x}'_{\text{even}}), \qquad \tilde{\boldsymbol{x}}_{\text{even}} = \boldsymbol{x}'_{\text{even}} \pm \eta(\boldsymbol{x}'_{\text{odd}}).$$

Re-interleave to a single stream:

$$p_{\theta_i}\big(\boldsymbol{x}_{\text{even}}, \boldsymbol{x}_{\text{odd}}\big) = \text{Interleave}(\tilde{\boldsymbol{x}}_{\text{even}}, \tilde{\boldsymbol{x}}_{\text{odd}}) \in \mathbb{R}^{L \times d}.$$

Apply $L$ levels recursively:

$$\hat{\boldsymbol{y}} = \big(p_{\theta_L} \circ g_L \circ \cdots \circ p_{\theta_1} \circ g_1\big)(\boldsymbol{x}).$$

**TimesNet (Wu et al., 2023).**

- **Choice of $K$.** $K = B$, the number of TimesBlocks.
- **Generator $g_i$ (frequency-to-time imaging).** Given $\boldsymbol{x} \in \mathbb{R}^{L \times d}$, compute $\widehat{\boldsymbol{x}} = \text{FFT}(\boldsymbol{x})$ and select $k$ dominant periods $\{p_m\}_{m=1}^k$. For each $p$, set $s = \lfloor L/p \rfloor$, $f = p$, and fold to a time–period image

$$\widetilde{\boldsymbol{x}}^{(p)} = \text{Reshape}(\boldsymbol{x}, s = \lfloor L/p \rfloor, f = p) \in \mathbb{R}^{d \times s \times f}, \qquad g_i(\boldsymbol{x}) = \{\widetilde{\boldsymbol{x}}^{(p_m)}\}_{m=1}^k.$$

- **Processor $p_{\theta_i}$ (2D conv mixing, adaptive aggregation).** Process each $\widetilde{\boldsymbol{x}}^{(p_m)}$ with a 2D block to obtain $\widehat{\boldsymbol{x}}_{2D}^{(p_m)}$, fold back to $\widehat{\boldsymbol{x}}_{1D}^{(p_m)} \in \mathbb{R}^{L \times d}$, and aggregate

$$Z_i = \sum_{m=1}^k \text{Conv2D}_{\theta_i}(\widetilde{\boldsymbol{x}}^{(p_m)}), \qquad \widehat{\boldsymbol{x}}_{1D}^{(p_m)} = \text{FoldBack}(\text{Proc}_{\theta_i}(\widetilde{\boldsymbol{x}}^{(p_m)})).$$

Let $A_{\ell-1}^{f_1}, \ldots, A_{\ell-1}^{f_k}$ be amplitudes from block $(\ell-1)$ and define

$$(\widetilde{A}_{\ell-1}^{f_1}, \ldots, \widetilde{A}_{\ell-1}^{f_k}) = \text{Softmax}(A_{\ell-1}^{f_1}, \ldots, A_{\ell-1}^{f_k}),$$

then

$$p_{\theta_i}(\{\widetilde{\boldsymbol{x}}^{(p_m)}\}_{m=1}^k) = \boldsymbol{x}_\ell^{1D} = \sum_{m=1}^k \widetilde{A}_{\ell-1}^{f_m} \odot \widehat{\boldsymbol{x}}_{1D}^{(p_m)} \in \mathbb{R}^{L \times d}.$$

**MICN (Wang et al., 2023).**

- **Choice of $K$.** $K = H$, the number of multi-branch inception stages.
- **Generator $g_i$ (downsampling, isometric convolution).** Given branch scale $K_i$ and current sequence $\boldsymbol{x}^{(i-1)} \in \mathbb{R}^{L_{i-1} \times d}$,

$$\boldsymbol{x}_{\text{gen}}^{(i)} = \text{Conv1D}(\text{AvgPool}_{K_i}(\boldsymbol{x}^{(i-1)}), \text{stride} = K_i) \in \mathbb{R}^{L_i \times d_i}, \qquad L_i = \left\lfloor \frac{L_{i-1}}{K_i} \right\rfloor,$$

and set $g_i(\boldsymbol{x}^{(i-1)}) = \boldsymbol{x}_{\text{gen}}^{(i)}$.
- **Processor $p_{\theta_i}$ (inception mixer: local and global fusion).** With local kernel set $\mathcal{R}$ and global branch $G_i$,

$$\text{LocalMix}_i(x) = \sum_{r \in \mathcal{R}} U_{i,r} \, \text{Conv1D}_{k=r}(x),$$

$$\text{GlobalMix}_i(x) = G_i\big(\text{Conv1D}_{\text{global}}(x)\big),$$

and the stage output

$$p_{\theta_i}(\boldsymbol{x}_{\text{gen}}^{(i)}) = \text{Merge}\big(\text{LocalMix}_i(\boldsymbol{x}_{\text{gen}}^{(i)}), \text{GlobalMix}_i(\boldsymbol{x}_{\text{gen}}^{(i)})\big) \in \mathbb{R}^{L_i \times d_i}.$$

**MSGNet (Cai et al., 2024).**

- **Choice of $K$.** $K = 1$.
- **Generator $g_1$ (Time imaging).** Let $\boldsymbol{x} \in \mathbb{R}^{N \times L \times d}$ (nodes $\times$ time $\times$ channels). Choose $\{p_m\}_{m=1}^M$ and set $s_m = \lfloor L/p_m \rfloor$, $f_m = p_m$. Reshape per node:

$$\widetilde{\boldsymbol{x}}^{(m)} = \text{Reshape}(\boldsymbol{x}, s_m, f_m) \in \mathbb{R}^{N \times d \times s_m \times f_m}, \qquad g_1(\boldsymbol{x}) = \{\widetilde{\boldsymbol{x}}^{(m)}\}_{m=1}^M.$$

- **Processor $p_{\theta_1}$ (multi-scale adaptive graph convolution and multi-head attention).** *(i) Graph filtering per scale.* Build a learnable adjacency

$$A^{(m)} = \text{Softmax}_{\text{row}}\big(\Phi_m(\text{Pool}_{t,f}(\widetilde{\boldsymbol{x}}^{(m)}))\big) \in \mathbb{R}^{N \times N}, \quad \tilde{L}^{(m)} = \text{I} - D^{(m)-\frac{1}{2}} A^{(m)} D^{(m)-\frac{1}{2}},$$

and apply a $K_g$-order Chebyshev filter

$$\mathcal{G}^{(m)} = \sum_{k=0}^{K_g} \theta_k^{(m)} T_k(\tilde{L}^{(m)}) \widetilde{\boldsymbol{x}}^{(m)} \in \mathbb{R}^{N \times d \times s_m \times f_m}.$$

*(ii) Multi-head attention over time–period tokens per scale.* Flatten $(s_m, f_m)$ to $S_m = s_m f_m$ tokens and let $Z^{(m)} \in \mathbb{R}^{(N \cdot d) \times S_m}$. For head $h$,

$$Q_h^{(m)} = Z^{(m)} W_{Q,h}^{(m)}, \quad K_h^{(m)} = Z^{(m)} W_{K,h}^{(m)}, \quad V_h^{(m)} = Z^{(m)} W_{V,h}^{(m)},$$

$$\mathrm{MHA}^{(m)}(Z^{(m)}) = \big[\mathrm{Concat}_h \, \mathrm{Softmax}(\tfrac{Q_h^{(m)} K_h^{(m)\top}}{\sqrt{d_k}}) V_h^{(m)}\big] W_O^{(m)}.$$

Unflatten and sum across scales:

$$Y^{(m)} = \mathrm{Unflatten}(\mathrm{MHA}^{(m)}(Z^{(m)}), s_m, f_m), \qquad \widehat{\boldsymbol{x}} = \sum_{m=1}^{M} \mathrm{FoldBack}(Y^{(m)}) \in \mathbb{R}^{N \times L \times d},$$

and predict

$$p_{\theta_1}(\{\widetilde{\boldsymbol{x}}^{(m)}\}_{m=1}^{M}) = \mathrm{TP}(\widehat{\boldsymbol{x}}) \in \mathbb{R}^{N \times T}.$$

**TimeMixer (Wang et al., 2024).**

- **Choice of $K$.** $K = 1$.
- **Generator $g_1$ (downsampling by average pooling).**

$$g_1(\boldsymbol{x}) = \{\boldsymbol{x}_m\}_{m=0}^{M}, \qquad \boldsymbol{x}_0 = \boldsymbol{x}, \quad \boldsymbol{x}_m = \mathrm{AvgPool}_{2^m}(\boldsymbol{x}) \in \mathbb{R}^{\lfloor L/2^m \rfloor \times d} \ (m \geq 1).$$

- **Processor $p_{\theta_1}$ (PDM$^L$, FMM).** Let $X^0 = \{\boldsymbol{x}_m\}_{m=0}^{M}$. For $\ell = 1, \ldots, L$ and each $m$,

$$(s_m^\ell, t_m^\ell) = \mathrm{Decompose}(X_m^{\ell-1}), \qquad X_m^{\ell-1} = s_m^\ell + t_m^\ell.$$

Bottom-up seasonal mixing $(s_{-1}^\ell = 0)$:

$$s_m^\ell \leftarrow s_m^\ell + \mathrm{BottomUpMixing}(s_{m-1}^\ell), \quad m = 0, \ldots, M.$$

Top-down trend mixing $(t_{M+1}^\ell = 0)$:

$$t_m^\ell \leftarrow t_m^\ell + \mathrm{TopDownMixing}(t_{m+1}^\ell), \quad m = M, \ldots, 0.$$

Scale-wise residual with cross-scale aggregation:

$$X^\ell = X^{\ell-1} + \mathrm{FFN}\big(\mathrm{SMix}(\{s_m^\ell\}_{m=0}^{M}) + \mathrm{TMix}(\{t_m^\ell\}_{m=0}^{M})\big).$$

Per-scale predictors and future multi-predictor mixing:

$$\hat{x}_m = \mathrm{Pred}_m(X_m^L) \in \mathbb{R}^T, \qquad p_{\theta_1}(X^0) = \mathrm{FMM}(\{\hat{x}_m\}_{m=0}^{M}) \in \mathbb{R}^T.$$

**TimeMixer++ (Wang et al., 2025).**

- **Choice of $K$.** $K = L$, which is the one generator–processor pair per MixerBlock.
- **Generator $g_i$ (recursive downsampling, multi-resolution time imaging).**

$$\boldsymbol{x}_m = \mathrm{Conv}(\boldsymbol{x}_{m-1}, \mathrm{stride} = 2) \in \mathbb{R}^{\lfloor T/2^m \rfloor \times C}, \ m = 1, \ldots, M, \qquad X_{\mathrm{init}} = \{\boldsymbol{x}_0, \ldots, \boldsymbol{x}_M\}.$$

*Input projection (channel mixing & embedding).*

$$\boldsymbol{x}_M \leftarrow \mathrm{Channel\text{-}Attn}(Q_M, K_M, V_M),$$

where

$$X^0 = \{\boldsymbol{x}_m^0\}_{m=0}^{M} = \mathrm{Embed}(X_{\mathrm{init}}), \ \boldsymbol{x}_m^0 \in \mathbb{R}^{\lfloor T/2^m \rfloor \times d_{\mathrm{model}}}.$$

*Per-block imaging.* From $X^{i-1} = \{\boldsymbol{x}_m^{i-1}\}$, estimate top-$K$ frequencies at the coarsest scale

$$A, \{f_k\}_{k=1}^{K}, \{p_k\}_{k=1}^{K} = \mathrm{FFT}(\boldsymbol{x}_M^{i-1}), \qquad p_k = \tfrac{T}{f_k},$$

and for each $m, k$,

$$z_m^{(i,k)} = \mathrm{Reshape}_{1D \to 2D}(\mathrm{Padding}_{m,k}(\boldsymbol{x}_m^{i-1})) \in \mathbb{R}^{p_k \times f_{m,k} \times d_{\mathrm{model}}}, \qquad f_{m,k} = \left\lceil \frac{\lfloor T/2^m \rfloor}{p_k} \right\rceil,$$

so

$$g_i(X^{i-1}) = \{z_m^{(i,k)} : m = 0, \ldots, M; \ k = 1, \ldots, K\}.$$

- **Processor** $p_{\theta_i}$ **(MixerBlock).**

$$X^i = \text{LayerNorm}\big(X^{i-1} + \text{MixerBlock}_{\theta_i}(g_i(X^{i-1}))\big),$$

with

$$(\text{TID}) \; s_m^{(i,k)} = \text{Attention}_{\text{col}}(\cdot), \quad t_m^{(i,k)} = \text{Attention}_{\text{row}}(\cdot),$$

$$(\text{MCM; seasonal bottom-up}) \; s_m^{(i,k)} \leftarrow s_m^{(i,k)} + \text{Conv2D}_{\downarrow 2}(s_{m-1}^{(i,k)}),$$

$$(\text{MCM; trend top-down}) \; t_m^{(i,k)} \leftarrow t_m^{(i,k)} + \text{TransConv2D}_{\uparrow 2}(t_{m+1}^{(i,k)}),$$

$$(\text{2D} \rightarrow \text{1D}) \; z_m^{(i,k)} \leftarrow \text{Reshape}_{\text{2D}\rightarrow\text{1D}}(s_m^{(i,k)} + t_m^{(i,k)}),$$

$$(\text{MRM; adaptive period fusion}) \; \widehat{A}^{f_k} = \text{Softmax}(A^{f_1}, \ldots, A^{f_K})_k, \quad \boldsymbol{x}_m^i = \sum_{k=1}^{K} \widehat{A}^{f_k} \odot z_m^{(i,k)},$$

- *Output projection.* With $X^L = \{\boldsymbol{x}_m^L\}_{m=0}^M$,

$$\widehat{y}_m = \text{Head}_m(\boldsymbol{x}_m^L) \in \mathbb{R}^T, \qquad \hat{\boldsymbol{y}} = \text{Ensemble}(\{\widehat{y}_m\}_{m=0}^M) \in \mathbb{R}^T.$$

**AMD (Hu et al., 2025a).**

- **Choice of** $K$. $K = 1$.
- **Generator** $g_1$ **(multi-scale moving-average, downsampling).** For $\boldsymbol{x} \in \mathbb{R}^T$ and rate $d \geq 2$,

$$\tau^{(1)} = \boldsymbol{x}, \qquad \tau^{(i)} = \text{AvgPool}_{k_i}(\tau^{(i-1)}) \in \mathbb{R}^{\lfloor T/d^{i-1} \rfloor}, \quad i = 2, \ldots, h,$$

and define $g_1(\boldsymbol{x}) = \{\tau^{(i)}\}_{i=1}^h$.
- **Processor** $p_{\theta_1}$ **(MDM, DDI, AMS).**
  *Coarse-to-fine residual mixing:*

$$\xi^{(h)} = \tau^{(h)}, \qquad \xi^{(i)} = \tau^{(i)} + \text{MLP}_i(\xi^{(i+1)}), \quad i = h-1, \ldots, 1,$$

and set $u = \xi^{(1)} \in \mathbb{R}^T$.
*DDI:* with patches $\widehat{U} = \mathcal{P}_P(u) \in \mathbb{R}^{n_P \times P}$,

$$Z = \widehat{U} + \text{MLP}_t(\widehat{U}), \qquad \widehat{V} = Z + \beta\big[\text{MLP}_c(Z^\top)\big]^\top,$$

then $v = \text{Unpatch}(\widehat{V}) \in \mathbb{R}^T$.
*AMS (dense MoE): selector*

$$S = \text{Softmax}(\text{TopK}(\text{Softmax}(Q(u)), k)),$$

$$Q(u) = \text{Decomp}(u) + \psi\,\text{Softplus}(\text{Decomp}(u)W_{\text{noise}}),$$

with $\psi \sim \mathcal{N}(0,1)$ and

$$\text{TopK}(z, k) = \begin{cases} \alpha\,\log(z+1), & z < v_k, \\ \alpha\,(e^z - 1), & z \geq v_k, \end{cases}$$

then

$$\hat{y} = \sum_{j=1}^{m} S_j\,\text{Predictor}_j(v) \in \mathbb{R}^T.$$

**WPMixer (Murad et al., 2025).**

- **Choice of** $K$. $K = 1$.
- **Generator** $g_1$ **(wavelet decomposition; time–frequency multiresolution).** Given $\boldsymbol{x} \in \mathbb{R}^{C \times L}$, perform an $m$-level DWT with mother wavelet $\psi$:

$$[\boldsymbol{x}_m^A, \boldsymbol{x}_m^D, \boldsymbol{x}_{m-1}^D, \ldots, \boldsymbol{x}_1^D] = \text{Decomp}(\boldsymbol{x}, \psi, m),$$

where $\boldsymbol{x}_i^A, \boldsymbol{x}_i^D \in \mathbb{R}^{C \times L_i}$ are approximation/detail series at level $i$. Retain

$$\mathcal{W} = \{\boldsymbol{x}_i^W\}_{i=1}^m, \qquad \boldsymbol{x}_m^W = \boldsymbol{x}_m^A, \; \boldsymbol{x}_i^W = \boldsymbol{x}_i^D \; (i < m), \qquad g_1(\boldsymbol{x}) = \{\boldsymbol{x}_1^W, \ldots, \boldsymbol{x}_m^W\}.$$

- **Processor $p_{\theta_1}$ per-resolution branch.** For each $i \in \{1, \ldots, m\}$:

$$\widetilde{\boldsymbol{x}}_i^W = \text{RevIN-Norm}(\boldsymbol{x}_i^W),$$

$$\boldsymbol{x}_i^P = \text{Patch}(\widetilde{\boldsymbol{x}}_i^W) \in \mathbb{R}^{C \times N_i \times P},$$

$$\boldsymbol{x}_i^d = \text{Embedding}(\boldsymbol{x}_i^P) \in \mathbb{R}^{C \times N_i \times d}.$$

*Patch Mixer:*

$$\boldsymbol{x}_i' = P(\text{BN}(\boldsymbol{x}_i^d)) \in \mathbb{R}^{d \times C \times N_i}, \qquad \boldsymbol{x}_i'' = \text{L}_2(\text{GELU}(\text{L}_1(\boldsymbol{x}_i'))).$$

*Embedding Mixer:*

$$\widehat{\boldsymbol{x}}_i = \text{BN}(P(\boldsymbol{x}_i'')) \in \mathbb{R}^{C \times N_i \times d}, \qquad Y_i^d = \widehat{\boldsymbol{x}}_i + \text{L}_2'(\text{GELU}(\text{L}_1'(\widehat{\boldsymbol{x}}_i))).$$

*Head:*

$$Y_i^f = \text{Flatten}(Y_i^d) \in \mathbb{R}^{C \times (N_i d)}, \qquad Y_i^h = \text{Linear}(Y_i^f) \in \mathbb{R}^{C \times T_i}.$$

*Reconstruction and denormalization:*

$$Y = \text{Reconstruction}_\psi(Y_m^A, Y_m^D, \ldots, Y_1^D) \in \mathbb{R}^{C \times T}, \qquad \widehat{\boldsymbol{x}} = \text{RevIN-Denorm}(Y) \in \mathbb{R}^{T \times C}.$$

**MultiPatchFormer (Naghashi et al., 2025).**

- **Choice of $K$.** $K = S$, which is the number of stages.
- **Generator $g_i$ (multi-scale patchifying).** Let $\boldsymbol{x}_{\text{in}} \in \mathbb{R}^L$. For scale $i \in \{1, \ldots, S\}$ choose $(P_i, S_i)$ and apply

$$E^{(i)} = \text{Conv1D}(\text{Pad}(\boldsymbol{x}_{\text{in}}), \text{ kernel} = P_i, \text{ stride} = S_i, \text{ out\_ch} = d_i) \in \mathbb{R}^{P_N \times d_i},$$

with scales chosen so $P_N$ is identical across $i$. Define

$$g_i(\boldsymbol{x}_{\text{in}}) = E^{(i)}, \qquad Z^{(i)} = \text{Concat}_d(Z^{(i-1)}, E^{(i)}) \in \mathbb{R}^{P_N \times d^{(i)}},$$

where $d^{(i)} = \sum_{j=1}^i d_j$.
- **Processor $p_{\theta_i}$.** Temporal self-attention over $P_N$ tokens:

$$Q_i = Z^{(i)} W_Q^{(i)}, \quad K_i = Z^{(i)} W_K^{(i)}, \quad V_i = Z^{(i)} W_V^{(i)} \in \mathbb{R}^{P_N \times d_h},$$

$$\text{MHA}_t(Z^{(i)}) = \left[ \text{Concat}_h \text{Softmax}\left(\frac{Q_i K_i^\top}{\sqrt{d_h}}\right) V_i \right] W_O^{(i)} \in \mathbb{R}^{P_N \times d^{(i)}}.$$

Fuse to width $d_*$ and apply LN+FFN:

$$U^{(i)} = \Phi_i(\text{Concat}_d(\text{MHA}_t(Z^{(i)}), H^{(i-1)})) \in \mathbb{R}^{P_N \times d_*}, \qquad H^{(i)} = \text{FFN}(\text{LN}(U^{(i)} + U^{(i)})).$$

Decoder at $i = S$: reduce to $z = \frac{1}{P_N} \sum_{p=1}^{P_N} H_{p,:}^{(S)}$ and, with $T = M\Delta$,

$$\hat{y}^{(1)} = \mathcal{D}_1(z) \in \mathbb{R}^\Delta, \qquad \hat{y}^{(m)} = \mathcal{D}_m(z, \text{Concat}(\hat{y}^{(1)}, \ldots, \hat{y}^{(m-1)})) \in \mathbb{R}^\Delta,$$

$$p_{\theta_i} = \begin{cases} (\text{MHA}_t \text{ \& Fuse \& LN+FFN}), & i < S, \\ (\text{MHA}_t \text{ \& Fuse \& LN+FFN \& MultiStepDecode}), & i = S. \end{cases}$$

In every case, $g_i$ yields a scale-specific view and $p_{\theta_i}$ performs learned processing within/across scales, establishing Definition 4.1.

$\square$

### A.4  PROOF OF THEOREM 5.2

*Proof.* We verify (i) *differentiability* in $\boldsymbol{s}$ and (ii) *consistency* to a (discrete) scaling operator family.

**Differentiability in $\boldsymbol{s}$.** For each fixed pair $(i, j)$, the mapping $s_i \mapsto e^{-s_i} I_{|i-j|}(s_i)$ is real-analytic on $\mathbb{R}_+$ because $e^{-s_i}$ and the modified Bessel function $I_\nu(s_i)$ are analytic for $s_i > 0$ and integer $\nu$. Hence $\boldsymbol{s} \mapsto \boldsymbol{K}(\boldsymbol{s})$ is $C^\infty$ on $\mathbb{R}_+^M$, and so is $\boldsymbol{s} \mapsto k(\boldsymbol{x}|\boldsymbol{s}) = \boldsymbol{K}(\boldsymbol{s})\boldsymbol{x}$ for any $\boldsymbol{x}$. This proves the differentiability requirement in Definition 5.1.

**Consistency to a discrete scaling operator family.** Fix $s \in \mathbb{Z}_+$ and set $\boldsymbol{s} = s\boldsymbol{1}$. Then $\boldsymbol{K}(s\boldsymbol{1})$ becomes space-invariant:

$$[\boldsymbol{K}(s\boldsymbol{1})]_{i,j} = e^{-s} I_{|i-j|}(s) =: g_s(i - j),$$

so $k(\boldsymbol{x}|s\boldsymbol{1})$ is the convolution of $\boldsymbol{x}$ with the kernel $g_s \in \ell^1(\mathbb{Z})$. Two classical properties hold:

*(a) Semigroup and normalization.* For all $s, t \geq 0$, $g_{s+t} = g_s * g_t$, and $\sum_{n \in \mathbb{Z}} g_s(n) = 1$ . Thus $\boldsymbol{K}(s\boldsymbol{1})$ is a symmetric, doubly-stochastic Markov operator.

*(b) Frequency response and contraction.* Let $\widehat{g}_s(\omega)$ denote the discrete-time Fourier transform (DTFT) of $g_s$. A standard computation gives

$$\widehat{g}_s(\omega) = \exp\big(s(\cos\omega - 1)\big) = \exp\big(-2s\sin^2(\omega/2)\big), \qquad \omega \in [-\pi, \pi],$$

hence $|\widehat{g}_s(\omega)| \leq 1$ with equality only at $\omega = 0$ for $s > 0$.

We now verify the two axioms in Definition 3.1 for the discrete family $f(\boldsymbol{x}|s) := k(\boldsymbol{x}|s\boldsymbol{1})$:

*Non-expansiveness.* By Plancherel,

$$\|f(\boldsymbol{x}|s) - f(\boldsymbol{x}'|s)\|_2^2 = \frac{1}{2\pi}\int_{-\pi}^{\pi} |\widehat{g}_s(\omega)|^2 \, |\widehat{\boldsymbol{x} - \boldsymbol{x}'}(\omega)|^2 \, d\omega \leq \|\boldsymbol{x} - \boldsymbol{x}'\|_2^2,$$

since $|\widehat{g}_s(\omega)| \leq 1$. Thus $\|f(\boldsymbol{x}|s) - f(\boldsymbol{x}'|s)\|_2 \leq \|\boldsymbol{x} - \boldsymbol{x}'\|_2$.

*Energy reduction.* Let $s_i, s_j \in \mathbb{Z}_+$ with $s_i = m s_j$ for some integer $m > 1$. Let $X(\omega)$ denote the DTFT of $\boldsymbol{x}$; then by Plancherel,

$$\|f(\boldsymbol{x}|s)\|_2^2 = \frac{1}{2\pi}\int_{-\pi}^{\pi} |\widehat{g}_s(\omega)|^2 \, |X(\omega)|^2 \, d\omega.$$

Using the semigroup property,

$$g_{s_i} = g_{s_j}^{*m} := \underbrace{g_{s_j} * \cdots * g_{s_j}}_{m \text{ times}},$$

the frequency responses satisfy

$$\widehat{g}_{s_i}(\omega) = \big(\widehat{g}_{s_j}(\omega)\big)^m, \qquad |\widehat{g}_{s_i}(\omega)|^2 = |\widehat{g}_{s_j}(\omega)|^{2m}.$$

Since $|\widehat{g}_{s_j}(\omega)| \leq 1$ for all $\omega$ and $m > 1$, we have

$$|\widehat{g}_{s_i}(\omega)|^2 = |\widehat{g}_{s_j}(\omega)|^{2m} \leq |\widehat{g}_{s_j}(\omega)|^2 \quad \text{for all } \omega.$$

Therefore, for any $\boldsymbol{x}$,

$$\|f(\boldsymbol{x}|s_i)\|_2^2 = \frac{1}{2\pi}\int_{-\pi}^{\pi} |\widehat{g}_{s_i}(\omega)|^2 \, |X(\omega)|^2 \, d\omega$$

$$\leq \frac{1}{2\pi}\int_{-\pi}^{\pi} |\widehat{g}_{s_j}(\omega)|^2 \, |X(\omega)|^2 \, d\omega$$

$$= \|f(\boldsymbol{x}|s_j)\|_2^2,$$

which proves the *energy reduction* inequality

$$\|f(\boldsymbol{x}|s_i)\|_2 \leq \|f(\boldsymbol{x}|s_j)\|_2 \quad \text{for all } \boldsymbol{x} \quad \text{whenever } s_i = m s_j, \ m > 1.$$

To see that the inequality is strict for some $\boldsymbol{x}$, note that for $s_j > 0$ we have

$$|\widehat{g}_{s_j}(\omega)| < 1 \quad \text{for all } \omega \neq 0,$$

and hence

$$|\widehat{g}_{s_i}(\omega)|^2 = |\widehat{g}_{s_j}(\omega)|^{2m} < |\widehat{g}_{s_j}(\omega)|^2 \quad \text{for all } \omega \neq 0.$$

Choose any $\boldsymbol{x}$ whose spectrum is not supported solely at $\omega = 0$, i.e., such that $|X(\omega)|^2 > 0$ on a set of nonzero measure in $\{\omega \in [-\pi, \pi] : \omega \neq 0\}$. Then on a set of positive measure,

$$|\widehat{g}_{s_i}(\omega)|^2 \, |X(\omega)|^2 < |\widehat{g}_{s_j}(\omega)|^2 \, |X(\omega)|^2,$$

so the inequality above is strict:

$$\|f(\boldsymbol{x}|s_i)\|_2 < \|f(\boldsymbol{x}|s_j)\|_2.$$

Thus the LDG kernel family satisfies both the universal inequality and the existence of a strict example required by the energy reduction condition in Definition 3.1.

$\square$

## A.5 PROOF OF THEOREM 5.3

### A.5.1 DISCRETE SCALE-SPACE AXIOMS

**Axioms (discrete, 1D, symmetric case).** For each scale $s \geq 0$, let $\mathcal{T}_s : \ell^2(\mathbb{Z}) \to \ell^2(\mathbb{Z})$ denote the smoothing operator at scale $s$. A family $\{\mathcal{T}_s\}_{s \geq 0}$ satisfies the *discrete scale-space axioms* if:

**(A1) Linearity and shift invariance.** $\mathcal{T}_s$ is linear and commutes with shifts: there exists $K_s \in \ell^1(\mathbb{Z})$ with $(\mathcal{T}_s x)[n] = (K_s * x)[n]$.

**(A2) Semigroup over scale and identity at** $0$. $\mathcal{T}_0 = \text{Id}$ and $\mathcal{T}_{s+t} = \mathcal{T}_s \circ \mathcal{T}_t$, i.e., $K_{s+t} = K_s * K_t$ for all $s, t \geq 0$.

**(A3) Regularity in scale.** The map $s \mapsto K_s$ is $C^1$ in $s$ in the $\ell^1$ topology.

**(A4) DC normalization.** $\sum_{n \in \mathbb{Z}} K_s[n] = 1$ for all $s \geq 0$ (constants are preserved).

**(A5) Non-enhancement of local extrema (NELE).** If $y(\cdot; s) = \mathcal{T}_s x$ has a local maximum at index $n$, then $\partial_s y[n; s] \leq 0$ (resp. $\partial_s y[n; s] \geq 0$ at local minima).

**(A6) Symmetry.** $K_s[n] = K_s[-n]$ for all $n \in \mathbb{Z}$ and $s \geq 0$.

### A.5.2 UNIQUENESS OF THE DISCRETE GAUSSIAN

**Lemma A.3.** *Let $A$ be a translation-invariant, symmetric operator on $\ell^2(\mathbb{Z})$ with impulse response $a[\cdot] \in \ell^1(\mathbb{Z})$ such that*

$$\sum_{n \in \mathbb{Z}} a[n] = 0, \qquad a[n] \geq 0 \text{ for } n \neq 0, \qquad a[-n] = a[n].$$

*Then its Fourier symbol $\phi(\omega) = \widehat{A}(\omega) = \sum_n a[n] e^{-in\omega}$ admits the Lévy-Khintchine form*

$$\phi(\omega) = \sum_{m=1}^{\infty} c_m \big( \cos(m\omega) - 1 \big), \qquad c_m \geq 0,$$

*with real coefficients $c_m = 2a[m]$. Conversely, given any sequence $\{c_m\}_{m \geq 1}$ with $c_m \geq 0$ and $\sum_{m \geq 1} c_m < \infty$, defining*

$$a[0] = -\sum_{m \geq 1} c_m, \qquad a[\pm m] = \tfrac{1}{2} c_m \ (m \geq 1),$$

*yields a symmetric Toeplitz operator $A$ with the above properties and symbol $\phi(\omega) = \sum_{m \geq 1} c_m (\cos m\omega - 1)$.*

*Proof.* Since $a \in \ell^1(\mathbb{Z})$ and $a[-n] = a[n]$,

$$\phi(\omega) = \sum_{n \in \mathbb{Z}} a[n] e^{-in\omega} = a[0] + 2 \sum_{m=1}^{\infty} a[m] \cos(m\omega).$$

The zero-sum condition gives $a[0] = -2 \sum_{m \geq 1} a[m]$, hence

$$\phi(\omega) = 2 \sum_{m=1}^{\infty} a[m] \big( \cos(m\omega) - 1 \big) = \sum_{m=1}^{\infty} c_m \big( \cos(m\omega) - 1 \big), \quad c_m := 2a[m] \geq 0.$$

Absolute summability of $a$ implies $\sum_m c_m < \infty$, ensuring uniform convergence and continuity of $\phi$. The converse follows by reversing the construction. $\square$

**Theorem A.4** (Uniqueness of the discrete Gaussian scale space)**.** *Among symmetric kernel families, the* discrete Gaussian

$$K_s[n] = e^{-\alpha s} I_{|n|}(\alpha s), \qquad \alpha > 0,$$

*is the unique extended scaling operator family that satisfies (A1)–(A6). Equivalently,*

$$\widehat{K}_s(\omega) = \exp\big( \alpha s(\cos \omega - 1) \big) = \exp\big( -2\alpha s \sin^2(\omega/2) \big), \qquad \omega \in [-\pi, \pi],$$

*and any other family satisfying the axioms coincides with $K_s$ up to reparameterizing scale $s \mapsto \alpha s$.*

*Proof.* By **(A1)**, $(\mathcal{T}_s x)[n] = (K_s * x)[n]$ for some $K_s \in \ell^1(\mathbb{Z})$. Let $\widehat{K}_s(\omega)$ be its DTFT. By **(A2)**–**(A3)**, for each fixed $\omega$ the map $s \mapsto \widehat{K}_s(\omega)$ is a continuous one-parameter semigroup with $\widehat{K}_0(\omega) = 1$, hence there exists a real, even *generator* $\phi(\omega)$ such that

$$\widehat{K}_s(\omega) = \exp\big(s\,\phi(\omega)\big), \qquad s \geq 0.$$

From **(A4)**, $\phi(0) = 0$. From **(A6)**, $\phi$ is real and even.

Let $y(\cdot\,; s) = \mathcal{T}_s x$. Differentiating in $s$,

$$\partial_s y = A * x, \qquad A := \partial_s K_s\big|_{s=0},$$

so that $\widehat{A}(\omega) = \phi(\omega)$. By **(A4)** and **(A6)**, $A$ is symmetric with zero row sum. The NELE axiom **(A5)** at vanishing scale enforces the discrete maximum principle for $A$: off-diagonal coefficients are nonnegative while the diagonal is nonpositive, and rows sum to zero. Thus $A$ fits the hypotheses of Lemma A.3, and

$$\phi(\omega) = \sum_{m=1}^{\infty} c_m\big(\cos(m\omega) - 1\big), \qquad c_m \geq 0.$$

We now show that **(A5)** forces $c_m = 0$ for all $m \geq 2$. Consider signals supported on three consecutive sites and apply **(A5)** at $s = 0$ to both local maxima and minima. A standard extremum test yields that any positive coefficient at distance $m \geq 2$ would produce, for sufficiently small $s > 0$, an increase at a newly formed off-center extremum before nearest neighbors equilibrate. Hence $c_m = 0$ for $m \geq 2$, and

$$\phi(\omega) = c_1(\cos\omega - 1) =: \alpha(\cos\omega - 1), \qquad \alpha := c_1 > 0.$$

Substituting into equation A.5.2 gives

$$\widehat{K}_s(\omega) = \exp\big(\alpha s(\cos\omega - 1)\big).$$

Taking the inverse DTFT yields $K_s[n] = e^{-\alpha s} I_{|n|}(\alpha s)$, the discrete Gaussian kernel. This family satisfies **(A1)**–**(A6)**; conversely, any other family obeying the axioms has the same generator up to the multiplicative constant $\alpha$, i.e., a reparameterization of scale. The result follows. $\qquad\square$

Table 5: Dataset descriptions for long-term and short-term forecasting benchmarks. The dataset size is organized as (Train, Validation, Test). For long-term tasks, we adopt multivariate datasets covering diverse domains. For short-term tasks, we follow the official M4 benchmark, which consists of univariate series at different temporal frequencies. Following TimeMixer++ (Wang et al., 2025), forecastability is measured as one minus spectral entropy (Goerg, 2013), with higher values indicating better predictability.

| Tasks | Dataset | Dim | Series Length | Dataset Size | Domain | Frequency | Forecast. |
|---|---|---|---|---|---|---|---|
| Forecasting (Long-term) | ETTm1 | 7 | {96, 192, 336, 720} | (34465, 11521, 11521) | Temperature | 15 min | 0.46 |
| | ETTm2 | 7 | {96, 192, 336, 720} | (34465, 11521, 11521) | Temperature | 15 min | 0.55 |
| | ETTh1 | 7 | {96, 192, 336, 720} | (8545, 2881, 2881) | Temperature | Hourly | 0.38 |
| | ETTh2 | 7 | {96, 192, 336, 720} | (8545, 2881, 2881) | Temperature | Hourly | 0.45 |
| | Electricity | 321 | {96, 192, 336, 720} | (18317, 2633, 5261) | Electricity | Hourly | 0.77 |
| | Traffic | 862 | {96, 192, 336, 720} | (12185, 1757, 3509) | Transportation | Hourly | 0.68 |
| | Exchange | 8 | {96, 192, 336, 720} | (5120, 665, 1422) | Exchange rate | Daily | 0.41 |
| | Weather | 21 | {96, 192, 336, 720} | (36792, 5271, 10540) | Weather | 10 min | 0.75 |
| Forecasting (Short-term) | M4-Yearly | 1 | 6 | (23000, 0, 23000) | Demographic | Yearly | 0.43 |
| | M4-Quarterly | 1 | 8 | (24000, 0, 24000) | Finance | Quarterly | 0.47 |
| | M4-Monthly | 1 | 18 | (48000, 0, 48000) | Industry | Monthly | 0.44 |
| | M4-Weekly | 1 | 13 | (359, 0, 359) | Macro | Weekly | 0.43 |
| | M4-Daily | 1 | 14 | (4227, 0, 4227) | Micro | Daily | 0.44 |
| | M4-Hourly | 1 | 48 | (414, 0, 414) | Other | Hourly | 0.46 |

Table 6: Experiment configurations of SIGMA across datasets. All experiments adopt the ADAM (Kinga et al., 2015) optimizer. We report the number of blocks ($K$), model dimension ($d_{\text{model}}$), initial learning rate (LR), loss function, batch size, and training epochs.

| Dataset | $K$ | $d_{\text{model}}$ | LR | Loss | Batch Size | Epochs |
|---|---|---|---|---|---|---|
| ETTh1 | 1 | 32 | 0.0005 | MSE | 32 | 10 |
| ETTh2 | 1 | 32 | 0.0005 | MSE | 32 | 10 |
| ETTm1 | 1 | 8 | 0.02 | MSE | 32 | 10 |
| ETTm2 | 1 | 16 | 0.0005 | MSE | 32 | 10 |
| Exchange | 1 | 16 | 0.0001 | MSE | 32 | 10 |
| Electricity | 1 | 16 | 0.01 | MSE | 32 | 10 |
| Traffic | 1 | 16 | 0.01 | MSE | 32 | 10 |
| Weather | 1 | 8 | 0.005 | MSE | 32 | 10 |
| M4 | 1 | 16 | 0.01 | SMAPE | 32 | 10 |

# B EXPERIMENTAL DETAILS

For the evaluation of forecasting models, we follow the protocol used in TimesNet Wu et al. (2023). For **long-term forecasting** (see Table 2), we report the mean square error (MSE) and mean absolute error (MAE). For **short-term forecasting** (see Table 3), we adopt symmetric mean absolute percentage error (SMAPE), mean absolute scaled error (MASE), and overall weighted average (OWA).

These metrics are defined as follows:

$$\text{SMAPE} = \frac{200}{H} \sum_{i=1}^{T} \frac{|\boldsymbol{X}_i - \hat{\boldsymbol{X}}_i|}{|\boldsymbol{X}_i| + |\hat{\boldsymbol{X}}_i|}, \tag{6}$$

$$\text{MASE} = \frac{1}{T} \sum_{i=1}^{T} \frac{|\boldsymbol{X}_i - \hat{\boldsymbol{X}}_i|}{\frac{1}{T-m} \sum_{j=m+1}^{T} |\boldsymbol{X}_j - \boldsymbol{X}_{j-m}|}, \tag{7}$$

$$\text{OWA} = \frac{1}{2} \left[ \frac{\text{SMAPE}}{\text{SMAPE}_{\text{Naive2}}} + \frac{\text{MASE}}{\text{MASE}_{\text{Naive2}}} \right], \tag{8}$$

where $m$ denotes the seasonal periodicity of the data, and $\boldsymbol{X} \in \mathbb{R}^{T \times C}$ and $\hat{\boldsymbol{X}} \in \mathbb{R}^{T \times C}$ represent the ground truth and predictions for $T$ future time steps with $C$ dimensions.

For methods that do not originally provide results on the M4 dataset, we rely on their official implementations and conduct a controlled hyperparameter search to ensure a fair comparison. All implementations are based on PyTorch (Paszke et al., 2019), with the modified Bessel function implemented via SciPy, and all experiments are executed on a single **NVIDIA RTX A6000** GPU. We summarize the datasets used in our experiments in Table 5, covering both long-term and short-term

Table 7: Standard deviation for SIGMA and the second-best method (AMD) on long-term forecasting datasets.

| Method | SIGMA | | AMD (2025a) | |
|---|---|---|---|---|
| Dataset | MSE | MAE | MSE | MAE |
| Weather | $0.247_{\pm 0.003}$ | $0.273_{\pm 0.002}$ | $0.283_{\pm 0.001}$ | $0.282_{\pm 0.001}$ |
| Electricity | $0.175_{\pm 0.001}$ | $0.269_{\pm 0.001}$ | $0.208_{\pm 0.000}$ | $0.303_{\pm 0.000}$ |
| Traffic | $0.458_{\pm 0.001}$ | $0.302_{\pm 0.001}$ | $0.546_{\pm 0.000}$ | $0.344_{\pm 0.000}$ |
| Exchange | $0.353_{\pm 0.001}$ | $0.400_{\pm 0.001}$ | $0.358_{\pm 0.009}$ | $0.401_{\pm 0.002}$ |
| ETTh1 | $0.443_{\pm 0.004}$ | $0.433_{\pm 0.002}$ | $0.447_{\pm 0.004}$ | $0.434_{\pm 0.003}$ |
| ETTh2 | $0.376_{\pm 0.003}$ | $0.402_{\pm 0.002}$ | $0.376_{\pm 0.022}$ | $0.400_{\pm 0.005}$ |
| ETTm1 | $0.383_{\pm 0.003}$ | $0.397_{\pm 0.002}$ | $0.395_{\pm 0.003}$ | $0.399_{\pm 0.001}$ |
| ETTm2 | $0.276_{\pm 0.001}$ | $0.322_{\pm 0.001}$ | $0.285_{\pm 0.043}$ | $0.328_{\pm 0.012}$ |

Table 8: Standard deviation for SIGMA and the second-best method (MultiPatchFormer) on the short-term forecasting dataset (M4).

| Method | SIGMA | | | MultiPatchFormer (2025) | | |
|---|---|---|---|---|---|---|
| Dataset | SMAPE | MASE | OWA | SMAPE | MASE | OWA |
| Yearly | $13.314_{\pm 0.022}$ | $2.989_{\pm 0.015}$ | $0.783_{\pm 0.002}$ | $13.296_{\pm 0.012}$ | $3.009_{\pm 0.008}$ | $0.785_{\pm 0.001}$ |
| Quarterly | $10.060_{\pm 0.052}$ | $1.177_{\pm 0.009}$ | $0.886_{\pm 0.005}$ | $10.166_{\pm 0.008}$ | $1.178_{\pm 0.003}$ | $0.892_{\pm 0.002}$ |
| Monthly | $12.750_{\pm 0.019}$ | $0.936_{\pm 0.006}$ | $0.882_{\pm 0.003}$ | $12.810_{\pm 0.016}$ | $0.942_{\pm 0.001}$ | $0.887_{\pm 0.001}$ |
| Others | $4.867_{\pm 0.046}$ | $3.316_{\pm 0.050}$ | $1.037_{\pm 0.011}$ | $4.849_{\pm 0.037}$ | $3.271_{\pm 0.023}$ | $1.028_{\pm 0.007}$ |
| Averaged | $11.840_{\pm 0.016}$ | $1.585_{\pm 0.009}$ | $0.868_{\pm 0.002}$ | $11.889_{\pm 0.010}$ | $1.591_{\pm 0.001}$ | $0.872_{\pm 0.000}$ |

forecasting tasks, and provide the detailed experimental settings, including model hyper-parameters and training setups, in Table 6.

## C  PROPERTIES OF SCALING OPERATOR FAMILY

Definition 3.1 characterizes a valid scaling operator through two essential properties: (1) non-expansiveness, requiring that applying the operator does not amplify differences between nearby inputs, and (2) energy reduction, requiring that coarser scales retain strictly less signal energy over multiplicable scales. These properties ensure that scaling progressively smooths the input while preserving stability.

Figure 6 reports the empirical behavior of all operator families in Table 1 across seven datasets. For every dataset, the induced output differences remain strictly smaller than the input differences, confirming non-expansiveness. Likewise, the average energy decreases monotonically over dyadic scales, indicating that each operator consistently removes high-frequency variation as the scale increases. Taken together, these results demonstrate that all operator families satisfy both conditions of Definition 3.1 not only in theory but also in practice, across diverse real-world time series.

## D  ERROR BARS

To assess the robustness of our experiments, we report the mean performance and standard deviation for SIGMA and the second-best baselines in Tables 7 and 8. On the long-term forecasting benchmarks, SIGMA achieves lower average MSE and MAE than AMD on nearly all datasets, while maintaining sufficiently small standard deviations.

On the short-term M4 benchmark, the averaged SMAPE, MASE, and OWA scores of SIGMA remain consistently better, and the corresponding standard deviations are of similar or smaller magnitude. Taken together, these error-bar analyses confirm that the gains reported by SIGMA are stable across runs and not attributable to random fluctuations.

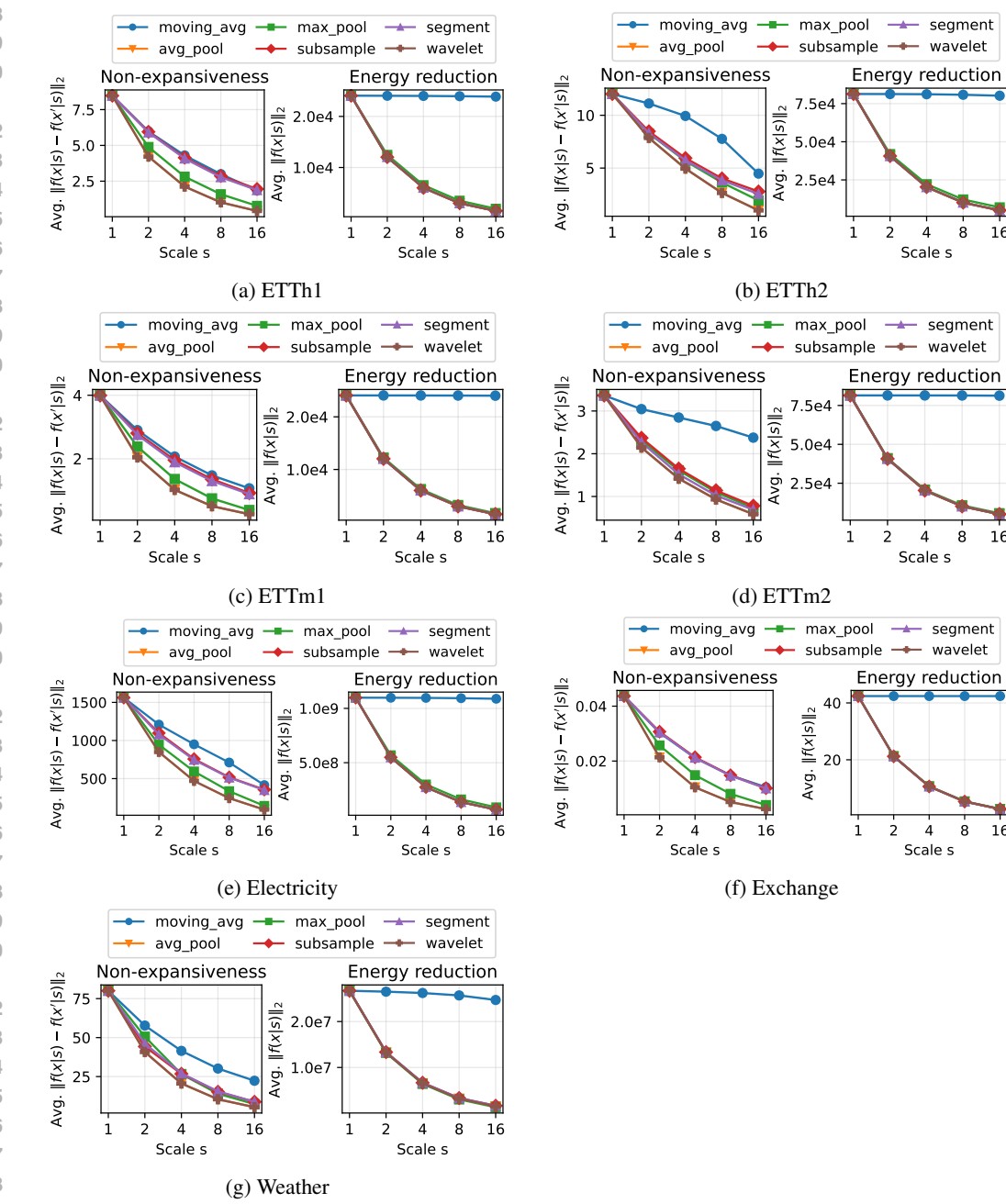

(a) ETTh1

(b) ETTh2

(c) ETTm1

(d) ETTm2

(e) Electricity

(f) Exchange

(g) Weather

Figure 6: Energy reduction and non-expansiveness of the scaling operator families across all datasets.

# E HYPERPARAMETER SENSITIVITY

Although we follow the evaluation protocol in TimesNet, the input length $L$ remains a key hyperparameter in multi-scale forecasting models. To understand its impact on both accuracy and efficiency, we measure forecasting performance (MSE) and computational overhead on ETTh1 under varying input lengths $L \in \{96, 192, 336, 720\}$. As shown in Figure 7, longer lookback windows generally lead to better forecasting performance, as the model can leverage richer historical patterns. We observe that the growth in both time and memory is approximately linear, indicating that the model scales efficiently with the window size.

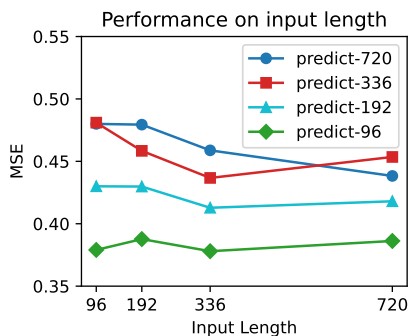 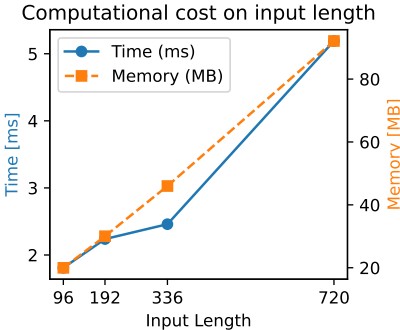

(a) Performance across input lengths on ETTh1. The overall trend shows improved MSE with longer lookback windows.

(b) Computational cost across of input lengths on ETTh1. Both metrics increase approximately linearly with respect to $L$.

Figure 7: Hyperparameter sensitivity with respect to the input length $L$ on ETTh1. We report forecasting performance (MSE) and computational cost to analyze the trade-off between accuracy and efficiency under varying lookback windows.

This efficiency arises because the LDG kernel is defined by a symmetric Toeplitz matrix whose entries depend only on the relative distance $|i - j|$. This structure implies that applying the LDG kernel to an input sequence is equivalent to performing a 1D convolution with a distance-dependent kernel vector, rather than multiplying by a full $L \times L$ matrix. Consequently, the operation can be computed in $O(L)$ time instead of $O(L^2)$, which explains why runtime and memory footprint grow approximately linearly with the input length.

## F  DESIGN GUIDE FOR EXTENDED SCALING OPERATOR FAMILIES

The extended scaling operator family proposed in Section 5.1 provides a flexible framework for constructing continuous, input-dependent multi-scale transformations. Here we summarize practical guidelines for designing new instances of this family and illustrate how these principles are instantiated in our implementation.

Classical downsampling operators are defined only at integer scales. To make these operators learnable and input-adaptive, we extend the discrete index $j \in \{1, \dots, J\}$ to a continuous scale variable $s_t \in [1, J]$ at each position $t$. Instead of selecting a single integer operator, we blend nearby integer scales through a convex combination

$$f(\boldsymbol{x}|s_t) = \sum_{j=1}^{J} w_j(s_t)\, f_j(\boldsymbol{x})_t,$$

where $f_j$ is the original operator at scale $j$. This convex construction preserves non-expansiveness whenever each $f_j$ is non-expansive, and it ensures that $f(\cdot|s)$ varies smoothly with respect to $s$.

To satisfy Definition 3.1, the extended operator must reproduce the original discrete operator exactly when $s_t$ is an integer. This is achieved by choosing weights $\{w_j(s)\}$ such that

$$w_j(k) = \mathbf{1}\{j = k\} \qquad \text{for all integers } k \in \{1, \dots, J\}.$$

A convenient construction uses $C^\infty$ bump functions:

$$w_j(s) = \frac{\phi(s - j)}{\sum_{k=1}^{J} \phi(s - k)}, \qquad \phi(u) = \begin{cases} \exp\big(-1/(1 - u^2)\big), & |u| < 1, \\ 0, & \text{otherwise.} \end{cases}$$

At integer $s = k$, the only nonzero bump is $\phi(0)$, which yields $w_k(k) = 1$ and $w_{j \neq k}(k) = 0$. Thus the extended operator reduces exactly to the classical operator at integer scales, ensuring consistency.

Because the bump function $\phi(\cdot)$ is $C^\infty$ and the normalization preserves smoothness, the weights $w_j(s)$ are differentiable in $s$ for all non-integer values. Since the output is a convex combination of the $f_j(\boldsymbol{x})$, the entire operator $f(\boldsymbol{x}|s)$ is differentiable in $s$, enabling backpropagation through the scale field and supporting input-adaptive scale prediction via a learnable scale head.

# G    COMPARISON WITH NON-MULTI-SCALE BASELINES

We evaluate SIGMA against strong non-multi-scale baselines, including TimeFilter (Hu et al., 2025b), PatchTST (Nie et al., 2023), and DLinear (Zeng et al., 2023) for long-term forecasting and short-term forecasting. Across the long-term forecasting benchmarks, SIGMA generally achieves the best accuracy on datasets with fewer variables (e.g., ETT). As the number of variables increases (e.g., Traffic), TimeFilter obtains the strongest overall performance, owing to its patch-wise filtration mechanism that explicitly captures spatiotemporal relationships. This suggests that explicitly modeling multi-scale patterns in multivariate settings can provide additional benefits. For short-term forecasting, SIGMA also performs strongly across all prediction horizons, demonstrating that SIGMA remains effective under diverse prediction-length settings.

Table 9: Long-term forecasting results with four methods: SIGMA, TimeFilter, PatchTST, and DLinear.

| Method | | SIGMA (Ours) | | TimeFilter (2025b) | | PatchTST (2023) | | DLinear (2023) | |
|---|---|---|---|---|---|---|---|---|---|
| Metric | | MSE | MAE | MSE | MAE | MSE | MAE | MSE | MAE |
| Weather | 96 | 0.160 | **0.204** | **0.159** | 0.205 | 0.174 | 0.216 | 0.196 | 0.258 |
| | 192 | 0.209 | **0.248** | **0.207** | 0.249 | 0.220 | 0.257 | 0.237 | 0.296 |
| | 336 | 0.270 | 0.293 | **0.264** | **0.291** | 0.278 | 0.298 | 0.283 | 0.333 |
| | 720 | 0.348 | 0.345 | **0.343** | **0.343** | 0.354 | 0.347 | 0.346 | 0.383 |
| | Avg | 0.247 | 0.273 | **0.243** | **0.272** | 0.256 | 0.279 | 0.266 | 0.318 |
| Electricity | 96 | 0.146 | 0.241 | **0.138** | **0.235** | 0.180 | 0.273 | 0.210 | 0.301 |
| | 192 | 0.163 | 0.257 | **0.156** | **0.253** | 0.187 | 0.279 | 0.210 | 0.305 |
| | 336 | 0.179 | 0.273 | **0.170** | **0.268** | 0.204 | 0.295 | 0.223 | 0.319 |
| | 720 | 0.213 | 0.304 | **0.191** | **0.289** | 0.246 | 0.328 | 0.258 | 0.350 |
| | Avg | 0.175 | 0.269 | **0.164** | **0.261** | 0.204 | 0.294 | 0.225 | 0.319 |
| Traffic | 96 | 0.431 | 0.288 | **0.391** | **0.260** | 0.460 | 0.298 | 0.696 | 0.429 |
| | 192 | 0.444 | 0.296 | **0.413** | **0.269** | 0.467 | 0.302 | 0.646 | 0.407 |
| | 336 | 0.461 | 0.303 | **0.429** | **0.277** | 0.483 | 0.308 | 0.653 | 0.410 |
| | 720 | 0.494 | 0.320 | **0.462** | **0.296** | 0.516 | 0.325 | 0.694 | 0.429 |
| | Avg | 0.458 | 0.302 | **0.424** | **0.276** | 0.482 | 0.308 | 0.672 | 0.419 |
| Exchange | 96 | 0.084 | 0.204 | **0.083** | **0.202** | 0.087 | 0.204 | 0.095 | 0.227 |
| | 192 | **0.174** | **0.297** | 0.178 | 0.299 | 0.188 | 0.308 | 0.184 | 0.323 |
| | 336 | **0.322** | **0.411** | 0.332 | 0.416 | 0.341 | 0.423 | 0.328 | 0.434 |
| | 720 | 0.833 | 0.687 | 0.785 | 0.669 | 0.921 | 0.720 | **0.762** | **0.667** |
| | Avg | 0.353 | 0.400 | 0.345 | **0.397** | 0.384 | 0.414 | **0.342** | 0.413 |
| ETTh1 | 96 | **0.379** | **0.393** | 0.384 | 0.395 | **0.379** | 0.398 | 0.396 | 0.410 |
| | 192 | **0.430** | 0.425 | 0.438 | **0.424** | **0.430** | 0.434 | 0.445 | 0.441 |
| | 336 | 0.481 | **0.446** | 0.481 | **0.446** | **0.473** | 0.460 | 0.493 | 0.471 |
| | 720 | 0.480 | 0.468 | **0.479** | **0.466** | 0.523 | 0.506 | 0.515 | 0.512 |
| | Avg | **0.443** | **0.433** | 0.446 | **0.433** | 0.451 | 0.450 | 0.462 | 0.459 |
| ETTh2 | 96 | **0.289** | **0.339** | 0.290 | 0.340 | 0.300 | 0.351 | 0.346 | 0.399 |
| | 192 | **0.369** | **0.394** | 0.377 | **0.394** | 0.380 | 0.400 | 0.478 | 0.477 |
| | 336 | **0.415** | **0.427** | 0.424 | 0.435 | 0.431 | 0.441 | 0.597 | 0.543 |
| | 720 | **0.431** | **0.446** | 0.464 | 0.464 | 0.447 | 0.461 | 0.841 | 0.661 |
| | Avg | **0.376** | **0.402** | 0.389 | 0.408 | 0.390 | 0.413 | 0.566 | 0.520 |
| ETTm1 | 96 | 0.323 | 0.359 | **0.320** | **0.357** | 0.330 | 0.368 | 0.345 | 0.372 |
| | 192 | **0.360** | **0.381** | 0.362 | **0.381** | 0.370 | 0.391 | 0.382 | 0.390 |
| | 336 | 0.392 | 0.404 | **0.391** | **0.402** | 0.402 | 0.411 | 0.414 | 0.414 |
| | 720 | **0.455** | 0.442 | 0.461 | **0.438** | 0.459 | 0.446 | 0.474 | 0.451 |
| | Avg | **0.383** | 0.397 | **0.383** | **0.394** | 0.390 | 0.404 | 0.404 | 0.407 |
| ETTm2 | 96 | 0.174 | **0.257** | **0.172** | 0.258 | 0.183 | 0.265 | 0.195 | 0.295 |
| | 192 | 0.239 | **0.299** | **0.237** | 0.300 | 0.246 | 0.308 | 0.282 | 0.359 |
| | 336 | **0.296** | **0.337** | 0.296 | 0.338 | 0.312 | 0.350 | 0.363 | 0.414 |
| | 720 | **0.394** | **0.394** | 0.394 | 0.396 | 0.419 | 0.412 | 0.547 | 0.519 |
| | Avg | 0.276 | **0.322** | **0.275** | 0.323 | 0.290 | 0.334 | 0.347 | 0.397 |

Table 10: Short-term forecasting results in the M4 benchmark dataset using four methods: SIGMA, TimeFilter, PatchTST, and DLinear.

| | Method | SIGMA (Ours) | TimeFilter (2025b) | PatchTST (2023) | DLinear (2023) |
|---|---|---|---|---|---|
| Yearly | SMAPE | **13.314** | 18.836 | 14.311 | 14.343 |
| | MASE | **2.989** | 4.153 | 3.240 | 3.123 |
| | OWA | **0.783** | 1.099 | 0.845 | 0.832 |
| Quarterly | SMAPE | **10.060** | 10.660 | 10.242 | 10.502 |
| | MASE | **1.177** | 1.228 | 1.211 | 1.240 |
| | OWA | **0.886** | 0.932 | 0.907 | 0.929 |
| Monthly | SMAPE | **12.750** | 13.477 | 12.889 | 13.373 |
| | MASE | **0.936** | 1.025 | 0.955 | 1.004 |
| | OWA | **0.882** | 0.949 | 0.896 | 0.935 |
| Others | SMAPE | 4.867 | 6.136 | **4.986** | 5.110 |
| | MASE | 3.316 | 4.042 | **3.231** | 3.655 |
| | OWA | 1.037 | 1.257 | **1.023** | 1.132 |
| Weighted Average | SMAPE | **11.840** | 13.666 | 12.186 | 12.494 |
| | MASE | **1.585** | 1.944 | 1.656 | 1.681 |
| | OWA | **0.868** | 0.995 | 0.893 | 0.920 |

