# OpenReview forum: "Unifying Multi-Scale Design in Time-Series Forecasting"
_ICLR.cc/2026/Conference — Submitted to ICLR 2026_

### Official Review · Reviewer_GCRB · 2025-10-15

**Soundness:** 3
**Presentation:** 3
**Contribution:** 3
**Rating:** 6
**Confidence:** 3

**Summary:**

This paper studies to unify scaling-related methods of TSF in one theory framework, based on which the authors also provide a simple and effective mutli-scale forecasting model. The authors conduct theory analysis to unify existing scaling operators and existing multi-scaling TSF methods as well.

**Strengths:**

1. Authors provide a unified definition and framework to define and study existing scaling operators. Based on the authors' definitions, they also study and unify the existing multi-scale-scalign-based time series forecasting methods through a framework. This is quite interesting and inspiring.
2. The proposed SIGMA method depends upon the proposed analyzing method. It makes sense to propose such method based on the authors' theory, and such method is shown to improve performance.

**Weaknesses:**

1. Though I'm not quite familiar with related math, but the proposed definition (i.e. definition 3.1) seems similar to a smooth multi-parameter scale-space/dilation family, followed by scale-dependent downsampling; "Non-expansiveness" equals being 1-Lipschitz, and "Entropy Reduction" equals a coarse-graining property. Moreover, it seems a part of theorem 3.2 (i.e. wavelet decomposition is scaling operator) relies on some assumptions stated in Appendix (i.e. Low-passdominance). Though I think these assumptions hold in most conditions, it's better to state them when proposing the theorem 3.2. Nevertheless, the definition seems clear and reasonable, not playing with difficult math concepts.

2. The training time comparison. AMD is a complicated framework, comparing training time agains which would seem a bit non-fair. Nevertheless, since SIGMA and AMD are all multi-scale framework, I think this is a small weakness.

**Questions:**

Please see weakness. Also:

Recent observations (e.g. https://www.arxiv.org/abs/2510.02729) show that these time series forecasting benchmarks have almost been saturated. Other concerns come from the fact that it doesn't seem to make sense that one simple neural network can reach sota for all time series tasks without any context (e.g. https://neurips.cc/virtual/2024/108471), so perhaps we should combine more things together (e.g. features, NNs, strategies, etc.). What's your opinion on these thoughts? Compared to proposing methods that risk overfitting these datasets, what future direction do you think would be good for further research of our time series community?

---

> ### Author Response · Authors · 2025-11-20
>
> We greatly appreciate your thoughtful review and constructive criticisms. Our detailed responses are provided below.
>
> ---
>
> **[W1] Clarifying the assumptions underlying Definition 3.1 and Theorem 3.2**
>
> We appreciate the reviewer’s point regarding the assumptions and the strength of the claims. In the revised version, we have revised and clarified our assumptions as follows:
>
> - **Dataset-level assumption**: We replace the previous low-pass dominance assumption with a minimal condition that the *forecastability* should be strictly below $ 1 $ (Section 2). Since the forecastability of $1$ represents that the whole time series collapses to a single frequency, almost all real-world datasets satisfy the assumption; we report the forecastability of all our datasets to confirm that this assumption holds in real-world settings (Appendix B, Table 5).
> - **Operator-level assumption**: We restate the key requirement as *energy reduction*, meaning that the $\ell_2$ norm of a time series decreases as the scale parameter increases multiplicatively, ensuring non-degenerate behavior (Section 3, Definition 3.1). We add a discussion clarifying how these conditions hold empirically across real-world datasets and how they relate to observable time-series characteristics. (Section 3, Figure 2; Appendix C, Figure 6)
> - **Theoretical guarantees**: We further clarify that all main theorems in our work continue to hold under our new assumption and requirement (Appendix A.1, A.2, A.4).
>
> ---
>
> **[W2] Interpreting the training time comparison between SiGMA and AMD**
>
> We agree that AMD is a sophisticated framework, and runtime comparisons must be interpreted with this complexity in mind. Our intention is similar to the positioning used in the DLinear paper, which compared linear models with Transformer-based architectures to illustrate the efficiency-accuracy trade-off across fundamentally different modeling paradigms. In the same spirit, our comparison emphasizes that strong accuracy within the multi-scale forecasting family can be achieved without incurring the substantial computational overhead associated with more elaborate designs. To ensure fairness, all models were trained under identical hardware, batch size, and optimization settings.

---

> > ### Author Response · Authors · 2025-11-20
> >
> > **[Q1] From saturated benchmarks to context-rich and principled modeling in time series community**
> >
> > Thank you for sharing the insightful observation. We agree that many standard time series forecasting benchmarks are becoming saturated, and that further small improvements on these datasets alone may have limited value for the field. We also share the concern that it is unlikely for a single, context-free neural architecture to be universally state-of-the-art across all time series tasks and domains.
> >
> > Our paper was written with this perspective in mind. Rather than starting from the goal of beating existing models on all benchmarks, we began from the question of how multi-scale temporal structure should be modeled in a principled way, and then used standard benchmarks primarily to verify that the resulting formulation is practically useful.
> > This is why we focus on formalizing a scaling operator family, analyzing properties such as non-expansiveness and energy reduction, and then designing the simplest architecture that can faithfully realize these principles, instead of combining many heterogeneous components or heavy modules. Our results should therefore be interpreted as evidence that a carefully designed multi-scale mechanism can be both simple and competitive on current benchmarks, not as a claim that one small network can solve all forecasting problems without context.
> >
> > In terms of future directions, we agree that simply stacking more features, neural networks, or strategies on top of saturated benchmarks is unlikely to be sufficient. We see two complementary directions as particularly important for the time-series community:
> > - **Better problem settings and benchmarks**: A natural next step is to design benchmarks that explicitly probe context, non-stationarity, and downstream decision constraints, rather than only average error on relatively narrow datasets. From a multi-scale viewpoint, this includes settings where interactions between short-term shocks and long-term trends, cross-resolution supervision, or regime changes localized at specific scales are central to the task. Evaluating scale-space generators under such conditions, possibly with rich covariates or contextual information, would clarify when principled multi-scale modeling provides benefits beyond current benchmarks.
> >
> > - **Modular, principled components**: In parallel, we believe it is important to develop building blocks with clear inductive biases that can be plugged into richer, context-aware systems. SiGMA is intended as one such module: its scaling operator framework can be extended with multivariate and cross-channel scaling, or with dynamic, context-dependent scale fields, and then combined with more expressive context encoders, graph or spatial modules. In this way, multi-scale generators can coexist with the ``more things together direction'' while still preserving interpretability and theoretical guarantees about how information flows across scales.
> >
> > Overall, our work is aligned with this perspective: SiGMA is designed as a principled, modular multi-scale component that performs well on existing benchmarks while being ready to be combined with richer context and more realistic problem settings in future research.
> >
> > ---
> >
> > We hope that these clarifications help to situate our contributions within the broader development of principled and interpretable time-series modeling. We would be pleased to incorporate any further refinements that the reviewers deem valuable.

---

> > > ### Comment · Reviewer_GCRB · 2025-11-21
> > >
> > > I would like to thank the authors for their timely reply. I believe they have resolved my concerns. **I would like to keep my rating as 6, while increase my confidences score to 4**.
> > >
> > > For the future direction, I think we indeed need better problem settings and benchmarks; while another thing I has been thinking about given your reply on modular and principled components is something like metalearning or paradigm researching. That is, we human researchers research for a "paradigm" or a method to obtain the best time series forecasting methods on better datasets. You see, like for stock forecasting, for weather forecasting, for electricity forecasting, etc., they have their own sub-field sotas. These sotas are not coming from our time series community, making it a bit meaningless to do ordinary research like what our community has been doing.
> > >
> > > Perhaps some future work could research on methods to select the best module or component (perhaps through searching algorithms or LLM-based methods) to do such meta-learning, etc.
> > >
> > > Afterall, I would like to call for researchers in time series forecasting community to go beyond building blocks targeting higher scores on existing benchmarks; but I think it could be better if we think of some more meaningful methods like meta-learning, etc. This comment is not that related to my up/down rating your paper, I am making this comment only because I think that this could be benefitial to our community and is related to your rebuttal reply.

---

> > > > ### Author Response · Authors · 2025-11-22
> > > >
> > > > We sincerely thank the reviewer for sharing these broader reflections on the future of the time-series forecasting community.
> > > >
> > > > We agree that current research directions increasingly rely not only on single end-to-end architectures, but also on higher-level procedures that can adaptively choose forecasting components across domains.
> > > > Several recent works support this view, showing that higher-level mechanisms can sit on top of modular forecasting components, for example through meta-learning or automated model selection [1,2].
> > > > In this context, our contribution is intended to fit naturally within this progression, as the reviewer suggested.
> > > >
> > > > At the same time, we also note that several recent studies argue that existing benchmarks can remain meaningful when used to study richer formulations of the forecasting problem.
> > > > For example, recent work explicitly analyzes temporal distribution shifts and concept drift in time-series forecasting, and proposes methods that treat these phenomena as central modeling and evaluation targets [3,4].
> > > > From this complementary perspective, benchmark datasets remain useful when the problem setting is broadened to include robustness to shifts, online adaptation, and the behavior under different regimes.
> > > >
> > > > We therefore see our work and the reviewer’s suggestions as aligned but operating at different levels: our theory aims to clarify and regularize one class of core modules (multi-scale scaling operators), while future research on meta-learning, paradigm design, and richer benchmark formulations can build on such modules to better address domain specificity, non-stationarity, and contextual information.
> > > >
> > > >
> > > > - [1] Rico Fischer and Ahmed Saadallah. AutoXPCR: Automated Multi-Objective Model Selection for Time Series Forecasting (KDD '24).
> > > > - [2] Weiqi Chen, Zhiqiang Zhu, Yiqi Zhang, Libin Shen, Linhui Yang, Qingsong Wen, and Liang Sun. Learning to Extrapolate and Adjust: Two-Stage Meta-Learning for Concept Drift in Online Time Series Forecasting (IJCAI '25).
> > > > - [3] Zheng Zhao, Huayi Liu, and B. Aditya Prakash. Tackling Time-Series Forecasting Generalization via Mitigating Concept Drift (arXiv '25).
> > > > - [4] Hyungjin Kim, Sangwoo Kim, Jiyoung Mok, and Sungroh Yoon. Battling the Non-Stationarity in Time Series Forecasting via Test-Time Adaptation (AAAI '25).

---

### Official Review · Reviewer_pAQg · 2025-10-30

**Soundness:** 3
**Presentation:** 3
**Contribution:** 3
**Rating:** 4
**Confidence:** 4

**Summary:**

This paper introduces a principled theoretical framework for multi-scale time-series forecasting by defining a novel "scaling operator family." Using this framework, the authors identify two key limitations in prior work: the use of non-adaptive, fixed scaling strategies and inflexible, staged cross-scale fusion mechanisms. To address these issues, they propose SIGMA (Single Gaussian Multi-scale Architecture), a simple yet powerful model. SIGMA's multi-scale generator is based on the learnable discrete Gaussian (LDG) kernel, which is grounded in scale-space theory and enables adaptive, data-dependent scaling. Its multi-scale processor is a lightweight MLP that efficiently fuses multi-resolution information in a single, flexible step. Comprehensive experiments on both long- and short-term forecasting benchmarks show that SIGMA achieves state-of-the-art accuracy, decisively outperforming recent, more complex models while being significantly more efficient in terms of training speed and memory consumption.

**Strengths:**

*   **Novety:** The paper's most significant contribution is the introduction of a formal framework for multi-scale modeling. By defining a "scaling operator family" through the properties of non-expansiveness and entropy reduction (Definition 3.1), the authors provide a much-needed theoretical foundation for a field previously driven by heuristics. The decomposition of multi-scale models into a "generator" and "processor" is an elegant abstraction that successfully unifies and clarifies the limitations of a wide range of existing methods (Theorem 4.2). This framework is a valuable and lasting contribution to the community.

*   **Theoretical Contribution:** The design of SIGMA is a direct and compelling instantiation of the paper's theoretical principles. The choice of the learnable discrete Gaussian (LDG) kernel as the generator is not ad-hoc; it is rigorously justified as the unique operator that satisfies the discrete scale-space axioms (Theorem 5.3). This provides a deep theoretical underpinning for the model's adaptive scaling capability. The architectural simplicity that follows—pairing the principled LDG kernel with a lightweight MLP processor—is a testament to the power of the framework, directly addressing the identified limitations of prior work.

*   **Empirical Performance:** The empirical validation is exceptionally strong and thorough. SIGMA demonstrates clear state-of-the-art performance across a wide array of long-term (Table 2) and short-term (Table 3) forecasting benchmarks. Achieving the best performance in 55 out of 80 long-term settings is a dominant result that is hard to dispute. The baselines are numerous, recent, and highly relevant, making the comparison robust and convincing.

*   **Computational Efficiency:** Beyond its superior accuracy, SIGMA's efficiency is a standout feature. The results presented in the efficiency analysis (Figure 3) are remarkable, showing improvements of up to 3.8x in memory and 5.3x in training speed over the strongest competitors. This is a rare combination of achieving both higher accuracy and higher efficiency, making SIGMA not only a theoretical but also a practical breakthrough. This advantage stems directly from the simplified, single-step architecture that avoids the sequential bottlenecks of staged fusion models.

**Weaknesses:**

In my opinion I found that authors have primarily concern the precise nature of the model's adaptivity and the scope of its components.

1.  **Limited Novelty of the "Processor" Component:** While the multi-scale generator (the LDG kernel) is highly novel and theoretically motivated, the multi-scale processor is a standard two-layer MLP. The innovation lies in the *simplicity* of the fusion process and the design of the inputs to the MLP (concatenating the smoothed and residual components), rather than in the processor's architecture itself. The paper's success suggests that with a sufficiently powerful and principled generator, an extremely simple processor is all that is needed, but the processor component itself is not a novel contribution.

2.  **Ambiguity in the Scope of "Adaptive Scaling":** The paper rightly claims its scaling is "adaptive" because the scale parameters `s` are learned from data, a clear advantage over fixed, hand-picked scales. However, the term "adaptive" could be interpreted in multiple ways. As presented, the scale parameters `s` appear to be part of the model's weights, learned during training and fixed at inference time. While they are specific to each time step's position within the input window, they do not dynamically adapt to the specific values of an incoming time series at test time. This is a more limited form of adaptivity than, for example, an attention mechanism that computes data-dependent weights for every new input.

3.  **Channel Independence as an Inherent Limitation:** The paper adopts a channel-independent approach, which is a common and effective strategy for ensuring applicability to multivariate settings and improving efficiency. However, this design choice inherently prevents the model from explicitly learning and leveraging cross-channel correlations at the multi-scale level. For datasets where such correlations are critical (e.g., the interactions between different sensors in a complex system), this could be a limitation compared to models that employ cross-channel mixing mechanisms.

**Questions:**

*   **Question 1:** The ablation study (Table 4) convincingly shows that using an invalid scaling operator like a standard convolution performs worst. From your theoretical perspective, which property of the scaling operator family (non-expansiveness or entropy reduction) do you believe is more critical for performance, and why does a standard learnable convolution fail to satisfy it?

*   **Question 2:** To clarify the nature of adaptivity: are the scale parameters `s ∈ R^L` learned as fixed model parameters during training, or are they dynamically computed based on the input sequence `x` at inference time? If they are fixed, have you considered a dynamic approach, and what would be the trade-offs?

*   **Question 3:** The performance gains are attributed to both the adaptive generator and the flexible processor. Could you provide an ablation that isolates their contributions? For instance, how would a model perform that uses the LDG kernel but with a more complex, staged processor from a prior work, or conversely, a model with fixed scaling (e.g., mean pooling) but with your simplified MLP processor?

*   **Question 4:** The paper demonstrates a single instantiation of the proposed generator-processor framework. Given the generality of your "scaling operator family," have you experimented with or can you speculate on other potential instantiations? For example, could a learnable wavelet transform that satisfies your definition serve as an alternative generator?

*   **Question 5:** The model's efficiency is a major advantage. However, the experiments use a fixed input length of L=96, which is shorter than what some state-of-the-art models use. How does SIGMA's performance and computational cost scale as the input lookback window `L` increases significantly (e.g., to 336 or 720)?

*   **Question 6:** Your framework's success with channel independence is impressive. In which specific real-world scenarios or dataset types do you foresee this assumption being a significant limitation, and how might your framework be extended to incorporate principled cross-channel interactions without sacrificing its core efficiency?

---

> ### Author Response · Authors · 2025-11-20
>
> Thank you for the constructive suggestions. We show below that the concerns can be resolved through additional analysis and clearer exposition.
>
> ---
>
> **[W1] Why a simple MLP processor is an intentional and desirable design choice in SiGMA**
>
> Thank you for the thoughtful comment.
> We believe that simplicity itself can be a source of novelty, as demonstrated in popular models such as DLinear, since simpler models are more preferable in practice. When the inductive bias provided by a principled multi-scale generator is correct, a lightweight processor becomes not only sufficient but also desirable for stability and efficiency.
>
> SiGMA intentionally adopts a simple MLP as the multi-scale processor. Once the multi-scale generator is mathematically well-structured, cross-scale fusion no longer requires a complex multi-stage architecture and can be implemented in a single step using a lightweight MLP. This one-shot design eliminates hand-crafted cross-scale pathways, addresses the inflexibility of existing multi-scale architectures, and makes the processor’s simplicity a natural part of the overall contribution rather than a lack of novelty.
>
>
> ---
>
> **[W2, Q2] Clarifying our use of "adaptive" scaling and discussing dynamic extensions**
>
> Thank you for the helpful suggestion. In the revised version, we clarify our terminology regarding “adaptive” scale parameterization. In response to Q2, the scale parameters are learned during training and remain fixed at inference. In other words, they are shared across all samples within the same dataset, but differ between datasets.
>
> We believe that a dynamic alternative represents a promising direction, consistent with our discussion in Section 7. A natural extension would be to retain the same generator structure but compute scale parameters in a data-dependent manner; for example, by predicting per-sample or per-position scale fields via an attention mechanism. This approach would allow the scales to vary with the input sequence while maintaining the theoretical structure of the LDG kernel family.
>
> While the dynamic approach increases expressiveness and sample-specific flexibility, it also introduces higher computational overhead, reduced interpretability of the scale field, and potentially more challenging stability constraints for maintaining the properties of scaling operator families (e.g., ordering). SiGMA currently prioritizes a stable and lightweight generator, and we view fully dynamic scaling as a promising direction for future development.
>
> ---
>
> **[W3, Q6] Scope and limitations of channel independence and potential cross-channel extensions**
>
> SiGMA intentionally focuses on adaptive temporal scaling, not cross-channel coupling.
> Our primary goal in this work is to isolate and understand the effect of principled temporal scaling, and to show that a well-structured multi-scale generator combined with a simple processor yields strong performance across a wide range of time series forecasting benchmarks.
>
> However, we agree that explicit multi-scale cross-channel fusion is valuable for domains with tightly coupled sensors. As mentioned in the conclusion, the incorporation of multivariate scaling mechanisms naturally emerges as a promising direction for future work. In particular, lightweight cross-channel fusion modules may complement SiGMA’s temporal scaling framework while maintaining its favorable robustness properties, and we consider this an important avenue for further exploration.

---

> > ### Author Response · Authors · 2025-11-20
> >
> > **[Q1] Roles of conditions in the scaling operator family and why standard convolutions fall short**
> >
> > Thank you for the insightful question. In the revised version, we explicitly formulate the operator requirements in terms of non-expansiveness and energy reduction.
> > The importance of each definition depends on the aspect of scaling one aims to preserve:
> >
> > - **Non-expansiveness** ensures *stability* : It guarantees that the $\ell_2$ norm does not increase across scales. If this condition is violated, small perturbations or high-frequency noise can be amplified when scaling operators are stacked, leading to growing activations, unstable multi-scale features, and error accumulation that is particularly harmful for long-horizon forecasting.
> >
> > - **Energy reduction** ensures a *coarse-to-fine hierarchy* : It requires that the $\ell_2$ norm decreases monotonically as the scale parameter increases multiplicatively.
> > If this condition is violated, coarser representations may reintroduce or even amplify local oscillations, breaking the intended coarse-to-fine structure.
> >   - **Difference between entropy and energy**:  We adopt energy reduction instead of entropy reduction in the revised version. The key difference is that energy reduction can be enforced at the operator level and, under a convex scaling family, holds for any input, whereas spectral entropy is strongly input-dependent and can increase for some signals even if the operator is intuitively smoothing.
> >
> > A standard learnable convolution does not satisfy these conditions because its weights are unconstrained: it has no explicit scale parameter $s$, no ordering over scales, and its operator norm can easily exceed $1$. In contrast, LDG kernels provide a normalized convolution that constitutes a valid scaling-operator family when the scale parameters are required to satisfy the appropriate ordering and boundedness conditions.
> >
> > ---
> >
> > **[Q3] Ablation study on multi-scale generator and multi-scale processor design**
> >
> > Thank you for the suggestion. We include an expanded ablation in the revised paper (Section 6, Table 4), and summarize the results in Table C below. The ablation examines the contribution of the LDG generator and the processor by evaluating the following variants:
> >
> > - (1) replacing the LDG kernel with a fixed moving-average operator,
> > - (2) replacing the LDG kernel with a mean-pooling operator, and
> > - (3) replacing the lightweight MLP processor with a more complex trend-seasonal mixing module inspired by TimeMixer.
> >
> > All experiments are conducted on ETTh1 with a prediction horizon of 720.
> > As shown in Table C, substituting the LDG kernel with fixed-scale operators ((1) and (2)) leads to a clear degradation in performance. In contrast, combining the LDG kernel with a more complex processor ((3)) yields performance comparable to SiGMA. These findings support our claim that the primary source of improvement lies in the adaptive generator, rather than in increasing processor complexity.
> >
> > **Table C. Ablation Summary**
> >
> > Best results are in **bold**, and second-best results are in *italics*.
> >
> >
> > | Method | MSE   | MAE   |
> > |--------|-------|-------|
> > | SiGMA  | **0.480** | *0.468* |
> > | (1)    | 0.493 | 0.475 |
> > | (2)    | 0.487 | 0.471 |
> > | (3)    | *0.486* | **0.467** |

---

> > > ### Author Response · Authors · 2025-11-20
> > >
> > > **[Q4] General design guide and example instantiations of extended scaling operator families**
> > >
> > > In the revised paper, we propose a general design guide for constructing extended scaling operator families (Appendix F). The core idea is to connect classical discrete operators through smooth differentiable weights, while preserving exact consistency at integer scales.
> > >
> > > For instance, we can construct an extended mean-pooling operator to a continuous
> > > scale field $s_t \in [1, J]$ by
> > > $$
> > > (MP(\mathbf{x}|\mathbf{s}))\_t
> > >   = \sum\_{j=1}^J w\_j(s\_t)(MP\_j \mathbf{x})\_t,
> > > $$
> > > where $MP_j$ denotes mean pooling with window size $j$.
> > > The weights are defined using a smooth bump-based cardinal construction:
> > >
> > > $$
> > > w\_j(s)
> > > = \frac{\phi(s - j)}{\sum_{k=1}^J \phi(s - k)},
> > > $$
> > > where $\phi(u)=-\tfrac{1}{1 - u^{2}}$ for $|u| < 1$ and $\phi(u)=0$ otherwise.
> > > At every integer scale $s = k$, the weight satisfies
> > > $w_k(k)=1$ and $w_{j\neq k}(k)=0$, ensuring exact reproduction of the discrete operator.
> > >
> > > We also implement this extended mean-pooling family on ETTh1 with predict-720 settings and report results in Table D. It shows that the extended family achieves performance competitive with SiGMA, which validates the proposed construction. These results indicate that the well-defined extended scaling operator family can match the performance of the LDG-based generator, while providing a complementary and conceptually simple example of our general design guide.
> > >
> > > **Table D. Extended mean-pooling operator family on ETTh1 (predict-720).**
> > >
> > >
> > > | Method         | MSE   | MAE   |
> > > |----------------|-------|-------|
> > > | SiGMA          | 0.480 | 0.468 |
> > > | ExtMeanPool | 0.478 | 0.466 |
> > >
> > >
> > > ---
> > >
> > > **[Q5] How SiGMA’s performance and efficiency scale with the input length $L$**
> > >
> > >
> > > We appreciate the reviewer’s concern regarding potential computational overhead. In fact, the LDG generator is highly efficient, yielding **linear** time and memory complexity in the input length $ L $. We include an additional empirical analysis in the revised paper (Appendix D).
> > >
> > > The linear complexity is made possible because the LDG kernel is symmetric and Toeplitz, with entries depending only on the relative distance $ |i - j| $.
> > > Applying the LDG operator is equivalent to performing a 1D convolution with a distance-dependent kernel, rather than performing a dense matrix multiplication.
> > > As shown in Figure 7 in the revised paper, both runtime and memory footprint increase approximately linearly with $ L $, demonstrating that the operator is scalable and numerically stable even for long sequences.
> > >
> > > ---
> > >
> > > We are grateful for the reviewer’s insightful comments, which have led to substantial improvements in the manuscript. We hope the revisions address all concerns and would be happy to provide further clarification if needed.

---

> > > > ### Comment · Reviewer_pAQg · 2025-11-26
> > > >
> > > > Thank you for your response.
> > > >
> > > > **Re: W1/Q3 (Novelty of MLP Processor and Ablation)**
> > > > The argument that a simple processor is an "intentional and desirable design choice" is noted. However, the new ablation study (Table C) is the critical piece of evidence. While it correctly shows that replacing the LDG generator with fixed-scale operators degrades performance, it *also* shows that replacing your simple MLP with a more complex processor from a prior work (TimeMixer) results in nearly identical performance (0.486 vs 0.480 MSE). This result actually weakens your claim. It suggests that once a powerful adaptive generator like the LDG is in place, the specific design of the processor—whether a simple MLP or a more complex module—is largely irrelevant. This finding undermines the claim that the "one-shot" MLP fusion is a key part of the contribution; rather, it appears that the generator is doing all the heavy lifting, and any reasonable processor suffices. This needs to be acknowledged.
> > > >
> > > > **Re: W2/Q2 (Scope of "Adaptive" Scaling)**
> > > > The clarification that the scale parameters `s` are learned per-dataset and are fixed at inference is crucial and was not clear from the original text. This is a much more limited form of "adaptivity" than the term typically implies in the context of sequence models (e.g., attention). It means the model adapts its scaling strategy to a new *distribution* (a dataset) but not to individual *instances* (sequences) at test time. The term "adaptive" is therefore potentially misleading. "Learnable" or "data-driven" scaling would be more precise. The significance of the method is constrained by this lack of instance-level adaptivity.
> > > >
> > > > **Re: W3/Q6 (Channel Independence)**
> > > > The argument that channel independence was an intentional choice to isolate temporal effects is valid from a research design perspective. However, it remains a significant practical limitation. By design, the model is incapable of learning cross-variable relationships, which are a primary driver of dynamics in many real-world multivariate systems (e.g., economics, climate science). While this simplifies the research question, it also narrows the applicability of the proposed model. This is not just a direction for future work; it is a fundamental architectural constraint that limits the scope of the current paper's claims about its general utility in multivariate forecasting.
> > > >
> > > > **Re: Q1/Q4 (Scaling Operator Families)**
> > > > The introduction of a general design guide for extended scaling operator families and the new "ExtMeanPool" experiment are strong additions. However, the fact that this newly constructed, conceptually simpler operator achieves performance *better* than the LDG kernel (0.478 vs 0.480 MSE) is a problematic result for the paper's narrative. It suggests that the specific choice of the theoretically elegant LDG kernel, which is presented as a cornerstone of the paper, is not actually necessary to achieve top performance. The core principle seems to be merely having *some* form of learnable, continuous scaling, rather than the specific scale-space-derived Gaussian form. This finding diminishes the claimed significance of the LDG kernel itself.
> > > >
> > > > **Re: Q5 (Scalability with Input Length)**
> > > > The confirmation that the LDG kernel is implemented as a convolution and thus scales linearly with sequence length `L` is a critical detail that should have been in the main paper. While this linear scaling is efficient, it does not match the O(L log L) complexity of state-of-the-art time-invariant SSMs. For extremely long sequences, where the primary motivation for SSMs lies, this linear complexity will eventually become a bottleneck compared to the quasi-linear scaling of convolution-based SSMs. Therefore, a trade-off exists: the proposed model gains the ability to handle non-stationarity at the cost of sacrificing some of the asymptotic efficiency that makes the SSM family attractive for very long sequence benchmarks. This trade-off is not discussed.
> > > >
> > > > **Conclusion:**
> > > > The response has clarified several key points, but the new evidence has also introduced new questions and weakened some of the original narrative's claims. The ablation in Table C and the new experiment in Table D collectively suggest that the LDG kernel is not uniquely effective and that the simple MLP processor is not a critical design choice. The limited scope of "adaptivity" and the inherent channel independence remain significant constraints. The paper presents a valuable framework for learnable scaling, but the specific instantiation (SIGMA) appears less uniquely principled than originally claimed.

---

> > > > > ### Author Response · Authors · 2025-11-28
> > > > >
> > > > > We thank the reviewer for the careful follow-up. We address each concern in detail below.
> > > > >
> > > > > ---
> > > > >
> > > > > **Re:Re: W1/Q3**.
> > > > >
> > > > > We agree that Table C indicates the LDG generator plays a central role in driving the accuracy gains.
> > > > > From this perspective, we do not view the result as a weakness, but as supporting evidence for our intended design: the generator is responsible for producing well-structured multi-scale representations, allowing the processor to remain simple without sacrificing performance.
> > > > >
> > > > > To clarify this point further, we conducted an additional efficiency experiment comparing two processor variants under the same LDG generator:
> > > > > - (1) the one-shot MLP processor,
> > > > > - (2) the multi-stage TimeMixer processor.
> > > > >
> > > > > As shown in Table E, the TimeMixer processor incurs notable computational overhead (10.4\% slower training and 14.8\% higher memory) without providing meaningful performance gains.
> > > > > These findings indicate that, given a well-structured multi-scale representation from the generator, a lightweight processor can function as an effective and efficient choice.
> > > > >
> > > > > **Table E. Efficiency comparison of multi-scale processors.**
> > > > > |Method|Training time (ms/iter)|Memory footprint (MB)|
> > > > > |-|-|-|
> > > > > |(1)|7.96|75.63|
> > > > > |(2)|8.79|86.84|
> > > > >
> > > > > ---
> > > > >
> > > > > **Re:Re: W2/Q2.**
> > > > >
> > > > > In the revision, we clarify that the scale parameters are learned during training and fixed at inference (line 364–365), and we use more precise terms such as "learnable" or "data-driven".
> > > > >
> > > > > We also agree that instance-level dynamic scaling is more closely aligned with the common use of the term "adaptive". Extending our framework in that direction is certainly promising, though it would require care to maintain the stability and interpretability that scaling operator families rely on. We see this as a meaningful direction for future work.

---

> > > > > > ### Author Response · Authors · 2025-11-28
> > > > > >
> > > > > > **Re:Re: W3/Q6.**
> > > > > >
> > > > > > We would like to clarify that channel independence (CI) is not a practical limitation of SiGMA, but rather reflects an intentional and empirically supported design choice. Recent analyses show that channel-independent models frequently outperform channel-dependent models because CI offers substantially higher robustness under distribution shift [1]. Consistent with these observations, our use of CI aims to isolate temporal modeling, reduce potential overfitting to unstable cross-channel relationships, and allow the model to adapt naturally when the set of variables changes.
> > > > > >
> > > > > > At the same time, we agree that cross-variable modeling can be beneficial when correlations are strong and stable. Extending our framework to incorporate principled cross-channel interactions is a meaningful direction for future work. Nonetheless, we do not believe that CI limits the practical applicability of the current model, and we hope our clarification helps convey the motivation behind this design choice.
> > > > > >
> > > > > > ---
> > > > > >
> > > > > > **Re:Re: Q1/Q4.**
> > > > > >
> > > > > > Our contribution lies in the formulation of the scaling operator family itself, rather than in the LDG kernel alone. In this sense, Table C supports the intended message of the paper.
> > > > > >
> > > > > > ExtMeanPool was deliberately constructed to fall within our extended scaling operator family, satisfying the same structural principles. Its strong performance is therefore consistent with our theoretical perspective: operators that adhere to these principles can be effective even when their functional form differs from that of LDG.
> > > > > >
> > > > > > It is also worth noting that not all forms of "learnable smoothing" are sufficient.
> > > > > > As shown in Table 4,
> > > > > > methods that do not satisfy these principles degrade significantly.
> > > > > > This underscores the role of the LDG kernel as a principled and well-characterized instance within the family and explains our choice to present it as the primary example. The ExtMeanPool result further demonstrates the flexibility of the framework and suggests room for even simpler or more efficient operators. We plan to expand these alternatives in the camera-ready version.
> > > > > >
> > > > > > ---
> > > > > >
> > > > > > **Re:Re: Q5.**
> > > > > >
> > > > > >
> > > > > > We would like to note that the linear-time complexity of SiGMA is an intentional consequence of its position-aware scaling design, and it reflects a modeling trade-off rather than an oversight.
> > > > > >
> > > > > > State-of-the-art SSMs achieve quasi-linear complexity by assuming time invariance, which yields a shift-invariant convolution kernel that can be accelerated with FFT-based methods [2].
> > > > > > In contrast, SiGMA learns position-dependent scales to capture non-stationary multi-scale structure, which inherently precludes FFT-based acceleration.
> > > > > >
> > > > > > We see this distinction as aligning with our primary focus: multi-scale forecasting methods rather than time-invariant SSMs. Within this multi-scale family, SiGMA matches or exceeds the efficiency of prior approaches while offering greater adaptivity.
> > > > > >
> > > > > > ---
> > > > > >
> > > > > > In summary, the additional analyses consistently reinforce the paper's main message: our theoretical framework characterizes what constitutes a valid scaling operator, and SiGMA is one principled instantiation within this family.
> > > > > > We believe the revision communicates this perspective more clearly and hope it conveys how the framework and SiGMA together offer a coherent, theoretically grounded approach to multi-scale forecasting.
> > > > > >
> > > > > > We sincerely appreciate the reviewer’s careful reading and thoughtful suggestions, which have helped us present the core ideas with greater clarity.
> > > > > >
> > > > > > - [1] Lu Han, Han-Jia Ye, and De-Chuan Zhan. The Capacity and Robustness Trade-off: Revisiting the Channel-Independent Strategy for Multivariate Time Series Forecasting (PVLDB '24).
> > > > > > - [2] Shriyank Somvanshi, Md Monzurul Islam, Mahmuda Sultana Mimi, Sazzad Bin Bashar Polock, Gaurab Chhetri, and Subasish Das. From S4 to Mamba: A Comprehensive Survey on Structured State Space Models (arXiv '25).

---

### Official Review · Reviewer_STwe · 2025-10-30

**Soundness:** 2
**Presentation:** 2
**Contribution:** 1
**Rating:** 2
**Confidence:** 4

**Summary:**

The paper introduces a unified theoretical framework for multi-scale modeling in time series forecasting, showing that most exisiting multi-scale approaches can fit into the framework.
The paper introduces a new multi-scale modeling method called SIGMA. Experiments show that SIGMA beats SOTA on both long-term and short-term tasks.

**Strengths:**

The theoretical framework looks strong, and the design of SIGMA has clear motivation. The experiment result also looks nice.

**Weaknesses:**

1. The theoretical framework, though looks solid, seems redundant to me. The two limitations, as well as the solutions proposed in SIGMA, look straightforward which you could easily arrive without the theoretical machinery. The theory feels **post-hoc** justification rather than guiding principle
2. The proof of theorem 3.2 in Appendix A proposes a new assumption, which is improper. Assumptions should be stated clearly in the main paper. Besides Assumption A.1 (Low-pass dominance) looks pretty strong, which may not hold for all real-world time series, especially those with strong periodic components or high-frequency dominant signals.
3. I don't think it is proper only comparing with multi-scale baselines as many models without explicit multi-scale designs naturally handle both long-term and short-term (e.g., PatchTST, DLinear variants).

There are some other minor issues related to their presentation, for example the standard deviation of three runs is not reported, etc.

**Questions:**

Can you provide more justification for Assumption A.1? Can you also compare the result with non-multi-scale baselines?

---

> ### Author Response · Authors · 2025-11-20
>
> We sincerely appreciate your constructive comments. Below, we provide detailed responses to each point you raised.
>
> ---
>
> **[W1] How the theoretical framework guides the design of SiGMA rather than serving as post-hoc justification**
>
> We appreciate the reviewer’s concerns regarding redundancy and potential post-hoc justification.
> However, we respectfully note that the limitations we discuss are not immediately obvious; they become clear only after these common properties are identified and their implications are analyzed.
>
> The resulting design choices in SiGMA, in particular the LDG kernel and the minimal single-stage processor, follow directly from these structural constraints rather than from heuristic reasoning. Without this theoretical grounding, the design space would be much broader, and the specific, coherent architecture we propose would not naturally emerge. Our ablation results further support this view: operators that violate the conditions in Definition 3.1 perform noticeably worse (Section 6, Table 4), indicating that the theoretically motivated constraints are closely tied to the effectiveness of the method.
>
> ---
>
> **[W2, Q1] Revising and justifying the assumptions underlying our theoretical analysis**
>
> We appreciate the reviewer’s point regarding the assumptions and the strength of the claims. In the revised version, we have revised and clarified our assumptions as follows:
>
> - **Dataset-level assumption**: We replace the previous low-pass dominance assumption with a minimal condition that the *forecastability* should be strictly below $1$ (Section 2). Since the forecastability of $1$ represents that the whole time series collapses to a single frequency, almost all real-world datasets satisfy the assumption; we report the forecastability of all our datasets to confirm that this assumption holds in real-world settings (Appendix B, Table 5).
> - **Operator-level assumption**: We restate the key requirement as *energy reduction*, meaning that the $\ell_2$ norm of a time series decreases as the scale parameter increases multiplicatively, ensuring non-degenerate behavior (Section 3, Definition 3.1). We add a discussion clarifying how these conditions hold empirically across real-world datasets and how they relate to observable time-series characteristics (Section 3, Figure 2; Appendix C, Figure 6).
> - **Theoretical guarantees**: We further clarify that all main theorems in our work continue to hold under our new assumption and requirement (Appendix A.1, A.2, A.4).
>
> **[W3, Q2] Evaluating SiGMA against non-multi-scale baselines for both long-term and short-term forecasting**
>
> Thank you for the suggestion. We conduct a comprehensive comparison with strong non-multi-scale baselines, including TimeFilter [1], PatchTST [2], and DLinear [3], for both long-term and short-term forecasting. The full results are provided in the revised paper (Appendix G), and we summarize the averages in Table A and Table B below.
>
> For long-term forecasting (Table A), SiGMA ranks first or second in 14 out of 16 configurations. It performs well on datasets with fewer variables (e.g., ETT), while TimeFilter performs best on high-dimensional datasets (e.g., Traffic). We view this gap as reflecting complementary modeling choices: SiGMA adopts a channel-independent design, whereas TimeFilter explicitly exploits cross-variable structure by its graph-based, channel-dependent architecture. In future work, we plan to investigate SiGMA variants with lightweight cross-channel or graph-based modules to better capture dependencies in high-dimensional multivariate settings.
>
> For short-term forecasting (Table B), SiGMA consistently outperforms all non-multi-scale baselines on the M4 benchmark, indicating that our approach remains effective under diverse prediction-length regimes.
>
> **Table A. Long-term forecasting results (average)**
> Best results are in **bold**, and second-best results are in *italics*.
>
> |Dataset|Metric|SiGMA (Ours)|TimeFilter|PatchTST|DLinear|
> |-|-|-|-|-|-|
> |Weather|MSE|*0.247*|**0.243**|0.256|0.266|
> ||MAE|*0.273*|**0.272**|0.279|0.318|
> |Electricity|MSE|*0.175*|**0.164**|0.204|0.225|
> ||MAE|*0.269*|**0.261**|0.294|0.319|
> |Traffic|MSE|*0.458*|**0.424**|0.482|0.672|
> ||MAE|*0.302*|**0.276**|0.308|0.419|
> |Exchange|MSE|0.353|*0.345*|0.384|**0.342**|
> ||MAE|*0.400*|**0.397**|0.414|0.413|
> |ETTh1|MSE|**0.443**|*0.446*|0.451|0.462|
> ||MAE|**0.433**|**0.433**|*0.450*|0.459|
> |ETTh2|MSE|**0.376**|*0.389*|0.390|0.566|
> ||MAE|**0.402**|*0.408*|0.413|0.520|
> |ETTm1|MSE|**0.383**|**0.383**|*0.390*|0.404|
> ||MAE|**0.397**|**0.394**|0.404|*0.407*|
> |ETTm2|MSE|*0.276*|**0.275**|0.290|0.347|
> ||MAE|**0.322**|*0.323*|0.334|0.397|
>
> **Table B. Short-term forecasting results (M4, weighted average)**
> Best results are in **bold**, and second-best results are in *italics*.
>
> |Metric|SiGMA (Ours)|TimeFilter|PatchTST|DLinear|
> |-|-|-|-|-|
> |SMAPE|**11.840**|13.666|*12.186*|12.494|
> |MASE|**1.585**|1.944|*1.656*|1.681|
> |OWA|**0.868**|0.995|*0.893*|0.920|

---

> > ### Author Response · Authors · 2025-11-20
> >
> > **There are some other minor issues related to their presentation, for example the standard deviation of three runs is not reported, etc.**
> >
> > We also include the standard deviation over three runs for both SiGMA and the second-best baseline in the revised version (Appendix D), following prior work [4,5]. Tables 7 and 8 in the revised paper show that SiGMA consistently exhibits small variance and is at least as stable as the strongest baselines, which supports the reliability of the reported improvements.
> >
> > ---
> >
> > We hope these clarifications address the emerging concerns and we are happy to make further refinements if needed.
> >
> > - [1] Yifan Hu, Guibin Zhang, Peiyuan Liu, Disen Lan, Naiqi Li, Dawei Cheng, Tao Dai, Shu-Tao Xia, and Shirui Pan. TimeFilter: Patch-specific spatial-temporal graph filtration for time series forecasting (ICML '25).
> > - [2] Yuqi Nie, Nam H Nguyen, Phanwadee Sinthong, and Jayant Kalagnanam. A Time Series is Worth 64 Words: Long-term forecasting with Transformers (ICLR '23).
> > - [3] Ailing Zeng, Muxi Chen, Lei Zhang, and Qiang Xu. Are Transformers effective for time series forecasting? (AAAI '23).
> > - [4] Shiyu Wang, Haixu Wu, Xiaoming Shi, Tengge Hu, Huakun Luo, Lintao Ma, James Y. Zhang, and Jun Zhou. TimeMixer: Decomposable multiscale mixing for time series forecasting (ICLR '24).
> > - [5] Shiyu Wang, Jiawei Li, Xiaoming Shi, Zhou Ye, Baichuan Mo, Wenze Lin, Shengtong Ju, Zhixuan Chu, and Ming Jin. TimeMixer++: A general time series pattern machine for universal predictive analysis (ICLR '25).

---

> ### Comment · Reviewer_STwe · 2025-11-20
>
> The additional experiment is good, especially with the convincing analysis why SIGMA can’t beat TimeFilter in some cases.
> The new assumptions look reasonable and clear. I will check details of the proof later.
>
> However, I’m still not convinced that the theory actually guides the design of the algorithm. For me, the two limitations directly come from observation of existing methods, and in order to get rid of the ‘fixed set of scales’ and improve ‘flexibility’, a natural way would be to use learnable scale parameters. I don’t see why we need the theory here. I would appreciate if the authors could state this more clearly, or point out exactly which part of the limitations or the algorithm is necessarily based on the theoretical framework or is otherwise very hard to come up with.
>
> But anyway, I would raise my score for the more comprehensive experiment result.

---

> > ### Author Response · Authors · 2025-11-22
> >
> > We thank the reviewer for the thoughtful follow-up and for raising the score.
> > We agree that “making scales learnable” is an intuitive direction to improve upon previous work, which can be studied even without our theoretical framework.
> >
> > Nevertheless, we believe that our theory guides *how to achieve it*, which involves several challenges that are difficult to address without clear guidance.
> > In the absence of a formal definition of scaling operator families and their extensions, learnable scales are under-specified and often degenerate into arbitrary smoothing; in particular, it is unclear in what sense a time series is *scaled*.
> > Multi-scale information in time series is useful when the scaling operators satisfy the following properties:
> >
> > - **Interpretability.** The scale parameter $s$ is interpretable in the continuous domain. In our framework, this is enforced by energy reduction: as the scale $s$ increases multicatively, the operator monotonically reduces the energy of the time series.
> >
> > - **Stability.** Representations remain stable across scales so that multi-scale modeling behaves predictably. In our framework, this is captured by non-expansiveness: the operator is required not to increase the distance between inputs across scales.
> >
> > Without these properties, we cannot guarantee that a learnable operator behaves as a valid scaling operator, even if it is parameterized by something called a scale. To make the above discussion concrete, we consider several naive designs that introduce learnable scales without theory:
> >
> > - **Independent per-scale modules.**
> >   Assume that each scale index is assigned an independent transformation. Since these modules are learned without any shared structural constraints, there is no meaningful way to interpret the role of the scale parameter.  Moreover, because no distance contraction is enforced, the operator may increase input-space distances across scales, leading to unstable multi-scale representations.
> >
> > - **Unnormalized convolution.**
> > The parameters of a convolution kernel are not interpretable as a scale, since changing them does not necessarily correspond to moving to a coarser representation. Moreover, without constraints such as non-expansiveness, a convolution operator can increase input-space distances arbitrarily, leading to unstable representations across scales and violating the expected behavior of a scaling operator.
> >
> >
> >
> > These examples illustrate that simply making scales learnable is not sufficient; enforcing structural properties is crucial for obtaining operators that  behave as scaling and for realizing a stable, interpretable multi-scale representation.

---

### Official Review · Reviewer_MsED · 2025-10-31

**Soundness:** 2
**Presentation:** 3
**Contribution:** 2
**Rating:** 4
**Confidence:** 3

**Summary:**

The paper builds a time-series forecaster from discrete scale-space theory. It adopts the discrete Gaussian (LDG) operator—a principled low-pass smoother with scale (s)—to create multi-scale representations (K_s * x) and fuses them with a small MLP for prediction. The goal is to combine interpretability (well-behaved smoothing, semigroup structure) with simplicity and efficiency. Experiments on standard benchmarks report competitive accuracy with a lightweight design.

**Strengths:**

The paper recasts multi-scale time-series modeling through discrete scale-space theory, arguing that any symmetric, linear, shift-invariant smoothing family satisfying standard axioms (semigroup, DC preservation, NELE) is uniquely the discrete Gaussian. It then builds a learnable LDG operator with position-adaptive scale $s_i$ , forms multi-scale representations, and fuses them with a small MLP to forecast. The authors claim: (i) principled, interpretable multi-scale smoothing; (ii) efficiency; (iii) competitive accuracy on common benchmarks.

**Weaknesses:**

1. The axiomatic uniqueness result is classical and sound, but the uniqueness guarantee does not necessarily transfer to the trained model.

2. The LDG kernel raises scalability concerns—position-adaptive scales imply $(O(L^2))$ time/space per sequence with nontrivial numerical stability, which can become prohibitive on long sequences.

3. Value lies in packaging a well-known prior (discrete Gaussian scale-space) into a light, learnable operator. Methodological originality is limited (multi-scale smoothing + small MLP). Practical impact is constrained by low-pass bias and narrow experiments.

4. Writing is generally clear, but strong phrases (“unique/principled”) are not tempered by discussion of assumption mismatches and failure cases (e.g., high-frequency-dominant series). The evaluation lacks high-pass comparisons, so the figures/ablations don’t clearly position the method.

**Questions:**

Does the “uniqueness” you prove actually show up in the trained model—not just in theory?

---

> ### Author Response · Authors · 2025-11-20
>
> We thank the reviewer for the thoughtful and constructive feedback. We address each point below.
>
> ---
>
> **[W1, Q1] How the axiomatic “uniqueness” property manifests in the trained model**
>
> We agree that the axiomatic uniqueness theorem characterizes the operator family rather than guaranteeing uniqueness of the learned instance. However, our model is explicitly constrained to remain within the uniquely defined discrete Gaussian scale-space family throughout training.
>
> The LDG kernel has a fixed analytical form uniquely determined by the axioms, and the model only learns the scale parameters $\mathbf{s} \in \mathbb{R}^L_{+}$. Thus, training selects a point inside the Gaussian semigroup rather than learning an arbitrary smoothing operator. In this sense, the learned model stays entirely within the theoretically guaranteed family, ensuring that all induced multi-scale representations remain principled and structurally consistent with the scale-space axioms.
>
> ---
>
> **[W2] Scalability and numerical stability of the LDG kernel on long input sequences**
>
> We appreciate the reviewer’s concern regarding potential computational overhead. In fact, the LDG generator is highly efficient, yielding **linear** time and memory complexity in the input length $ L $. We include an additional empirical analysis in the revised paper (Appendix D).
>
> The linear complexity is made possible because the LDG kernel is symmetric and Toeplitz, with entries depending only on the relative distance $ |i - j| $.
> Applying the LDG operator is equivalent to performing a 1D convolution with a distance-dependent kernel, rather than performing a dense matrix multiplication.
> As shown in Figure 7 in the revised paper, both runtime and memory footprint increase approximately linearly with $ L $, demonstrating that the operator is scalable and numerically stable even for long sequences.
>
> ---
>
> **[W3] How the proposed framework delivers methodological novelty and practical impact**
>
> We would like to clarify the contributions of our work more explicitly:
>
> - **Theoretical and conceptual contribution**:
> We develop the first unified mathematical framework for multi-scale modeling in time series.
> Building on this foundation, we introduce the novel concept of an extended scaling operator family, which enables the scaling to be learnable.
> This framework clarifies what fundamentally constitutes a legitimate multi-scale transformation under a common theory and provides a rigorous basis for designing new architectures.
>
> - **Methodological originality**:
> We intentionally adopt a simple, efficient MLP as the multi-scale processor. Once the multi-scale generator is mathematically well-structured, cross-scale fusion no longer requires a complex staged mechanism. This design also addresses the limitations of existing multi-scale models that rely on rigid two-stage fusion pipelines.
>
> - **Practical impact**:
> We evaluate SiGMA against 8 state-of-the-art multi-scale and 3 non-multi-scale baselines on 8 standard long-term multivariate benchmarks, each with 4 prediction horizons spanning energy, transportation, finance, and weather, as well as on the large-scale M4 short-term benchmark covering 6 temporal frequencies.
> We report an efficiency analysis, a qualitative case study of learned scales, extended ablations, and empirical verification that our theoretical assumptions are satisfied on all our datasets.
>
> ---
>
> **[W4] Assumptions, limitations, and potential failure cases of the proposed multi-scale framework**
>
> We appreciate the reviewer’s point regarding the assumptions and the strength of the claims. In the revised version, we have revised and clarified our assumptions as follows:
>
> - **Dataset-level assumption**: We replace the previous low-pass dominance assumption with a minimal condition that the *forecastability* should be strictly below $ 1 $ (Section 2). Since the forecastability of $1$ represents that the whole time series collapses to a single frequency, almost all real-world datasets satisfy the assumption; we report the forecastability of all our datasets to confirm that this assumption holds in real-world settings (Appendix B, Table 5).
> - **Operator-level assumption**: We restate the key requirement as *energy reduction*, meaning that the $\ell_2$ norm of a time series decreases as the scale parameter increases multiplicatively, ensuring non-degenerate behavior (Section 3, Definition 3.1). We add a discussion clarifying how these conditions hold empirically across real-world datasets and how they relate to observable time-series characteristics. (Section 3, Figure 2; Appendix C, Figure 6)
> - **Theoretical guarantees**: We further clarify that all main theorems in our work continue to hold under our new assumption and requirement (Appendix A.1, A.2, A.4).
>
> ---
>
> We thank the reviewer again for the insightful comments and welcome further discussion if additional clarification would be helpful.

---

> > ### Comment · Reviewer_MsED · 2025-11-25
> >
> > Thanks for authors' clarifications. I am willing to raise my rating to 6.

---

> > > ### Author Response · Authors · 2025-11-25
> > >
> > > We sincerely thank the reviewer for the careful rereading and for raising the rating. We appreciate the constructive engagement throughout the discussion and are pleased that our clarifications have addressed the reviewer’s concerns. Please let us know if any additional points would benefit from further explanation.

---

### Author Response · Authors · 2025-11-20
**Overall response to all reviewers**

We sincerely thank all reviewers for their detailed, constructive, and insightful feedback. Your comments have substantially improved both the theoretical clarity and empirical validation of our work. Below, we summarize the main changes in the revised manuscript.

---

## Textual revisions

- **Assumption validity and theoretical scope (MsED, STwE, GCRB)**:
  - Replaced the low-pass dominance assumption with a dataset-level forecastability condition with empirical validation. *[Section 2; Appendix B (Table 5)]*
  - Reformulated the operator-level requirement of entropy reduction into energy reduction with empirical analysis. *[Section 3 (Definition 3.1, Figure 2); Appendix C (Figure 6)]*
  - Clarified that all theorems remain valid under the mild conditions.  *[Appendix A.1–A.4]*


- **Clarifying "adaptive" scaling terminology (pAQg)**:
  Stated clearly that position-wise scales are learned per dataset and fixed at inference. *[Lines 16, 18, 48–49, 73–75, 92, 253, 267, 282, 285, 301, 364–365, 423, 446, 463, 504]*

- **Design guide for extended scaling operator families (pAQg)**: Added a general guideline for constructing extended scaling operator families. *[Appendix F]*


---

## Experimental additions

- **Ablation on generator and processor (pAQg)**: Added ablations that replace the lightweight MLP multi-scale processor with a more complex one. *[Table 4]*

- **Error bars (STwe)**: Reported the standard deviation over three runs for SiGMA and the second-best baselines.  *[Appendix D]*

- **Hyperparameter sensitivity on input length (MsED, pAQg)**:  Analyzed how performance and computational cost change across different input lengths. *[Appendix E]*

- **Comparison with non-multi-scale baselines (STwe)**:  Added experiments comparing SiGMA with non-multi-scale baselines for both long-term and short-term forecasting. *[Appendix G]*



---

Thank you again for your careful evaluation. Any further feedback would be very welcome and valuable for improving this work.

---

### Author Response · Authors · 2025-12-02
**General Response to Area Chairs**

Dear Area Chairs,

We sincerely thank you for the time and effort dedicated to ensuring a fair and thoughtful evaluation of our submission. Below, we provide a brief summary of how the revised manuscript and our responses address the key concerns raised during the review.

- **Limited scope of dataset assumption (MsED, STwe, pAQg, GCRB).** Reviewers noted that our dataset-level assumption was too restrictive.
In response, we:
  - (1) replaced low-pass dominance with an empirically validated forecastability condition (Section 2),
  - (2) reformulated entropy reduction as energy reduction (Section 3),
  - (3) confirmed that all theoretical results remain valid (Appendix A).

- **Insufficient verification of the theoretical framework (MsED, STwe, pAQg).** Reviewers requested clearer evidence that the theory meaningfully guides the model design. We showed that:
  - (1) operators violating our axioms degrade notably (Table C),
  - (2) the axioms explain failure cases of naive learnable scaling,
  - (3) the learned model remains within the Gaussian family.

- **Lack of complexity analysis (MsED, pAQg).** We added theoretical and empirical analyses demonstrating that the LDG generator has linear time and memory complexity (Appendix E).

- **Missing comparisons with non-multi-scale baselines (STwe).** We added experiments with PatchTST, DLinear, and TimeFilter, and SiGMA remains competitive or superior in most settings (Appendix G, Table C, Table D).

- **Lack of error bars (STwe).** We added standard deviations over three runs for SiGMA and strong baselines (Appendix D).

- **Ambiguity of the term "adaptive" (pAQg).** We clarified that scales are learned per dataset, fixed at inference, and revised wording to "learnable" or "data-driven" scaling.

- **Limitation from channel independence (pAQg)**. We clarified that channel independence is an intentional robustness-oriented choice, while noting that principled cross-channel extensions naturally fit within our framework.

- **Insufficient ablation of generator vs. processor (pAQg).** We added ablations showing that the generator is the main performance driver, and that more complex processors incur overhead without meaningful gains (Table 4, Table E).

- **Necessity of alternative generators (pAQg).** We introduced a general guide for constructing extended scaling operator families and implemented an extended mean-pooling operator (ExtMeanPool) that fits our theory and performs competitively (Appendix F, Table D).

- **Concerns regarding benchmark saturation (GCRB).** We clarified that our goal is to provide a principled, modular multi-scale component that can be integrated into richer benchmarks, context-aware settings, and higher-level frameworks.

All reviewers acknowledged that our clarifications addressed their concerns, with several updates:
- Reviewer MsED increased the rating from 4 to 6,
- Reviewer STwe increased the rating from 2 to 4 and the contribution score from 1 to 2,
- Reviewer GCRB raised the confidence score from 3 to 4 while maintaining rating 6.

We hope these updates indicate that our revisions and additional analyses were helpful in clarifying the points raised in the initial reviews. We are grateful for the constructive engagement throughout the process and for the opportunity to clarify the contribution of this work.



Thank you again for your careful reading and consideration.

Warm regards,

The Authors

---

### Meta-Review · Area_Chair_xFVt · 2026-01-07

**Summary:**

This paper proposes a theoretical framework for multi-scale time-series forecasting and instantiates it with a learnable scaling model, named SiGMA, based on a discrete Gaussian generator and a simple MLP processor.

Reviewers acknowledged the clear motivation of the framework design, theoretical contribution, empirical performance, and computational efficiency.  During the rebuttal, the authors have improved this paper by refining the dataset assumption, providing more verification of the theoretical framework, adding theoretical and empirical analyses, reporting error bars, clarifying the channel independence, providing more ablation of generator vs. processor, introducing a general guide for constructing extended scaling operator families, and implementing an extended mean-pooling operator (ExtMeanPool).

However, the newly added experimental results weaken the original claim of this paper, as pointed out by Reviewer pAQg. Specifically, the new ablation study (Table C) correctly shows that replacing the LDG generator with fixed-scale operators degrades performance. Besides, it also shows that replacing the simple MLP with a more complex processor from a prior work (TimeMixer) results in nearly identical performance (0.486 vs 0.480 MSE).  As acknowledged by the authors, the LDG generator plays a central role in driving the accuracy gains. Therefore, the necessity and distinctiveness of the specific SiGMA instantiation become unclear, and the contribution appears closer to demonstrating that learnable continuous scaling is beneficial, rather than that the proposed design is uniquely principled. Specifically, the link between the theoretical framework and the framework design remains insufficiently compelling.

In summary, while the paper is thoughtful and much improved, I lean towards rejection due to remaining concerns about positioning, necessity, and clarity of the core contribution. I encourage the authors to resubmit after repositioning the contributions.

**Reviewer Concerns:**

**The Addressed Concerns**
Concerns about empirical completeness and basic technical soundness raised by reviewers MsED, STwe, and GCRB were largely resolved via:
+ The authors clarified and weakened the original assumptions (replacing low-pass dominance with a forecastability condition and reformulating entropy reduction as energy reduction)
+ The authors added missing experimental evidence (non-multi-scale baselines, error bars over multiple runs)
+ The authors provided both theoretical and empirical analyses of computational complexity and scalability.

**The Remained Concerns**
+ It is still unclear whether the proposed theoretical framework genuinely guides algorithmic design rather than serving as a post-hoc formalization.
+ The scope of “adaptive” scaling is limited to per-dataset learned parameters fixed at inference, and the channel-independent design imposes a nontrivial restriction on applicability that is not fully reflected in the paper’s claims.

**Reviewer Scores:**

+ Reviewer MsED would like to increase the rating from 4 to 6,
+ Reviewer STwe would like to increase the rating from 2 to 4 and the contribution score from 1 to 2.
+ Reviewer GCRB would like to raise the confidence score from 3 to 4 while maintaining the rating of 6.
+ Reviewer pAQg would like to keep his/her rating.

---

### Decision · Program_Chairs · 2026-01-26

Reject